# Bacterial encapsulins as orthogonal compartments for mammalian cell engineering

Felix Sigmund[1,2,3], Christoph Massner[1,2,3], Philipp Erdmann[4], Anja Stelzl[1,2], Hannes Rolbieski[1,2], Mitul Desai[5], Sarah Bricault[5], Tobias P. Wörner[6], Joost Snijder[6,7], Arie Geerlof[8], Helmut Fuchs [9], Martin Hrabĕ de Angelis [9], Albert J.R. Heck [6], Alan Jasanoff[5,10,11], Vasilis Ntziachristos[1,12], Jürgen Plitzko[4] & Gil G. Westmeyer [1,2,3]

We genetically controlled compartmentalization in eukaryotic cells by heterologous expression of bacterial encapsulin shell and cargo proteins to engineer enclosed enzymatic reactions and size-constrained metal biomineralization. The shell protein (*EncA*) from *Myxococcus xanthus* auto-assembles into nanocompartments inside mammalian cells to which sets of native (*EncB,C,D*) and engineered cargo proteins self-target enabling localized bimolecular fluorescence and enzyme complementation. Encapsulation of the enzyme tyrosinase leads to the confinement of toxic melanin production for robust detection via multispectral optoacoustic tomography (MSOT). Co-expression of ferritin-like native cargo (*EncB,C*) results in efficient iron sequestration producing substantial contrast by magnetic resonance imaging (MRI) and allowing for magnetic cell sorting. The monodisperse, spherical, and iron-loading nanoshells are also excellent genetically encoded reporters for electron microscopy (EM). In general, eukaryotically expressed encapsulins enable cellular engineering of spatially confined multicomponent processes with versatile applications in multiscale molecular imaging, as well as intriguing implications for metabolic engineering and cellular therapy.

[1] Institute of Biological and Medical Imaging, Helmholtz Zentrum München, Ingolstädter Landstraße 1, Neuherberg 85764, Germany. [2] Institute of Developmental Genetics, Helmholtz Zentrum München, Ingolstädter Landstraße 1, Neuherberg 85764, Germany. [3] Department of Nuclear Medicine, Technical University of Munich, Ismaninger Straße 22, Munich 81675, Germany. [4] Department of Structural Biology, Max Planck Institute of Biochemistry, Am Klopferspitz 18, Martinsried 82152, Germany. [5] Department of Biological Engineering, Massachusetts Institute of Technology, 77 Massachusetts Avenue, Cambridge 02139 Massachusetts, USA. [6] Biomolecular Mass Spectrometry and Proteomics Group, Bijvoet Center for Biomolecular Research and Utrecht Institute for Pharmaceutical Sciences, Utrecht University, Padualaan 8, Utrecht 3584CH, The Netherlands. [7] Snijder Bioscience, Spijkerstraat 114-4, Arnhem 6828 DN, The Netherlands. [8] Institute of Structural Biology, Helmholtz Zentrum München, Ingolstädter Landstraße 1, Neuherberg 85764, Germany. [9] Institute of Experimental Genetics, Helmholtz Zentrum München, Ingolstädter Landstraße 1, Neuherberg 85764, Germany. [10] Department of Brain & Cognitive Sciences, Massachusetts Institute of Technology, 77 Massachusetts Avenue, Cambridge 02139 Massachusetts, USA. [11] Department of Nuclear Science & Engineering, Massachusetts Institute of Technology, 77 Massachusetts Avenue, Cambridge 02139 Massachusetts, USA. [12] Chair for Biological Imaging, Technical University of Munich, Ismaninger Straße 22, Munich 81675, Germany. Correspondence and requests for materials should be addressed to G.G.W. (email: gil.westmeyer@tum.de)

Compartmentalization, the spatial separation of processes into closed subspaces, is an important principle that has evolved on several biological scales: multi-enzyme complexes that channel substrates, nanocompartments built entirely from proteins, as well as membrane-enclosed organelles, cells, and organs. Compartments make it possible to generate and maintain specific local conditions that can facilitate interactions and reactions in confined environments[1], such that they can isolate toxic reaction products, protect labile intermediate products from degradation, or separate anabolic from catabolic processes[2]. Whereas eukaryotes possess many membrane-enclosed organelles, membranous compartments are not known in bacteria with a notable exception of magnetosomes in magnetotactic bacteria, in which specific reaction conditions are maintained which enable magnetic biomineralization[3,4]. However, nanocompartment shells built entirely from protein complexes can serve functions in prokaryotes that are analogous to eukaryotic organelles[5].

Intense work has been invested in engineering compartments in prokaryotic systems and yeast to realize features such as substrate channeling for biotechnological production processes[1,6,7]. In contrast, no orthogonal compartments with self-targeting cargo molecules exist to date for use in mammalian cells. Such a system could, for instance, enable cellular engineering of reaction chambers that would endow genetically modified mammalian cells with new metabolic pathways that may include labile intermediate products or spatially confined toxic compounds. Engineered orthogonal compartments in eukaryotic cells may also enable size-constrained synthesis of biomaterials via, e.g., metal biomineralization processes occurring under specific localized environmental conditions.

With regards to protein complexes as building blocks for addressable nanocompartments, viruses and virus-like particles have been expressed in bacterial hosts to encase fluorescent proteins[8–12], enzymes[13–16], and even multi-enzymatic processes[17,18]. Similarly, bacterial microcompartments (BMC) such as Eut microcompartments and carboxysomes have been genetically engineered to load foreign cargo proteins such as fluorescent proteins[19–21]. In mammalian systems, vault proteins (vaults) have been explored, which are ribonucleoprotein complexes enclosed by ~60 nm large envelope structures[22] into which foreign cargo proteins such as fluorescent proteins or enzymes can be packaged[23–25]. However, vaults have openings on both ends and are endogenously expressed by many eukaryotic cells[26]. With respect to protein shell structures that can incorporate iron, the iron storage protein ferritin has been overexpressed to generate MRI contrast under certain conditions although its core size is only ~6 nm containing only ~2000 iron atoms on average per core, which can result in only low magnetization[27–29]. Viral capsids such as the ones from cowpea chlorotic mottle virus (CCMV) have also been equipped with iron-binding sites that lead to accumulation of iron. Expression, assembly, and iron loading in mammalian cells, however, have not yet been demonstrated[30].

In search of a versatile nanocompartment-cargo system for heterologous expression in eukaryotic cells, we were intrigued by the recently discovered class of prokaryotic proteinaceous shell proteins called encapsulins because they possess a set of attractive features: (1) A single shell protein—without the need for proteolytic processing—is sufficient to form comparably large shell-like architectures (~18 nm or ~32 nm) auto-assembled from 60, or 180 identical subunits with a triangulation number ($T$) of one ($T = 1$) or three ($T = 3$), respectively[31]. (2) The assembled shells are pH resistant and temperature stable[32]. (3) A versatile set of native cargo molecules including enzymes exist that are packaged into shell structures via specific encapsulation signals defined by a short terminal peptide sequence[32]. (4) The pore size of ~5 Å allows channeling small molecular substrates through the shell[33]. (5) *Myxococcus xanthus* (*M. xanthus*) encapsulin was also shown to posses cargo proteins B and C, both containing rubrerythrin/ferritin-like domains as well as highly conserved iron-binding ExxH motifs[31] enabling import and sequestration of iron inside the nanoshell. Ferritin-like cargo proteins adopt an "open ferritin structure" and possess ferroxidase activity[31,33]. A model based on structural data from *Thermotoga maritima* encapsulins (with $T = 1$) assumes that the ferritin-like protein docks into the shell where it obtains ferrous iron through the pores which it then oxidizes for deposition of up to an estimated 30,000 iron atoms per shell in the case of *M. xanthus*[31,33,34]. This amount is an order of magnitude more than can be contained inside a ferritin core expressed in eukaryotic cells. The function of the cargo protein D, on the other hand, has so far not been understood. (6) The termini of the shell protein extend to the inner and outer surface, respectively, such that surface functionalizations are conveniently possible. The outer surface can, for instance, be functionalized to install specific targeting moieties[35–37]. Encapsulin variants were also purified when secreted from HEK293 cells to present glycosylated epitopes for an innovative vaccination approach[38]. Recently, the inner surface of the shell from *T. maritima* was also modified with silver-binding peptides to cause local silver precipitation in *Escherichia coli*, but the ferritin-like cargo was deleted to achieve this feature[39]. (7) Non-native cargo proteins including enzymes can be addressed to the inside of the nanocompartment via a short encapsulation signal[11,40].

This excellent set of studies showed the feasibility and utility of biotechnological production of encapsulins as biomolecular scaffolds and targetable vehicles and probes.

We here introduce engineered encapsulins modified from *M. xanthus* in the context of genetic programming of orthogonal and addressable cellular compartments in mammalian cells. We demonstrate that eukaryotically expressed encapsulins not only auto-assemble at high density and without toxic effects but that self-targeting and encapsulation of cargo molecules still efficiently occur in mammalian cells. We furthermore show localized enzymatic reactions in the nanocompartment useful for optical and optoacoustic imaging, as well as confined iron accumulation within the nanocompartments that labels cells for detection by MRI. Importantly, we also show that encapsulins can serve as excellent gene reporters for electron microscopy due their spherical shape and their ability to load iron. These data demonstrate the value of encapsulins as genetic markers across modalities. In addition, the iron sequestration inside the nanoshells affords magnetic manipulation of cells genetically labeled with encapsulins.

## Results

**Encapsulin expression and self-assembly**. Based on the favorable set of features introduced above, we chose to heterologously overexpress the encapsulin shell protein from *M. xanthus* in HEK293T cells. We tagged the nanoshell with an outward facing FLAG epitope (A$^{FLAG}$) and found it to express strongly without and with the native cargo molecules from *M. xanthus*, denoted encapsulins B, C, and D[31].

Co-expression of Myc-tagged B, C, or D alone, or a combination of all three non-tagged proteins (via co-transfection or a P2A construct, Fig. 1b), co-immunoprecipitated with A$^{FLAG}$ as visualized on silver-stained SDS-PAGE (Fig. 1c, middle panel). A corresponding western blot against the FLAG (Fig. 1c, upper panel) or Myc-epitope (Fig. 1c, lower panel) confirmed the identities of the protein bands (A$^{FLAG}$: 32.9 kDa, $^{Myc}$B: 18.5 kDa, $^{Myc}$C: 15.4 kDa, $^{Myc}$D: 12.5 kDa).

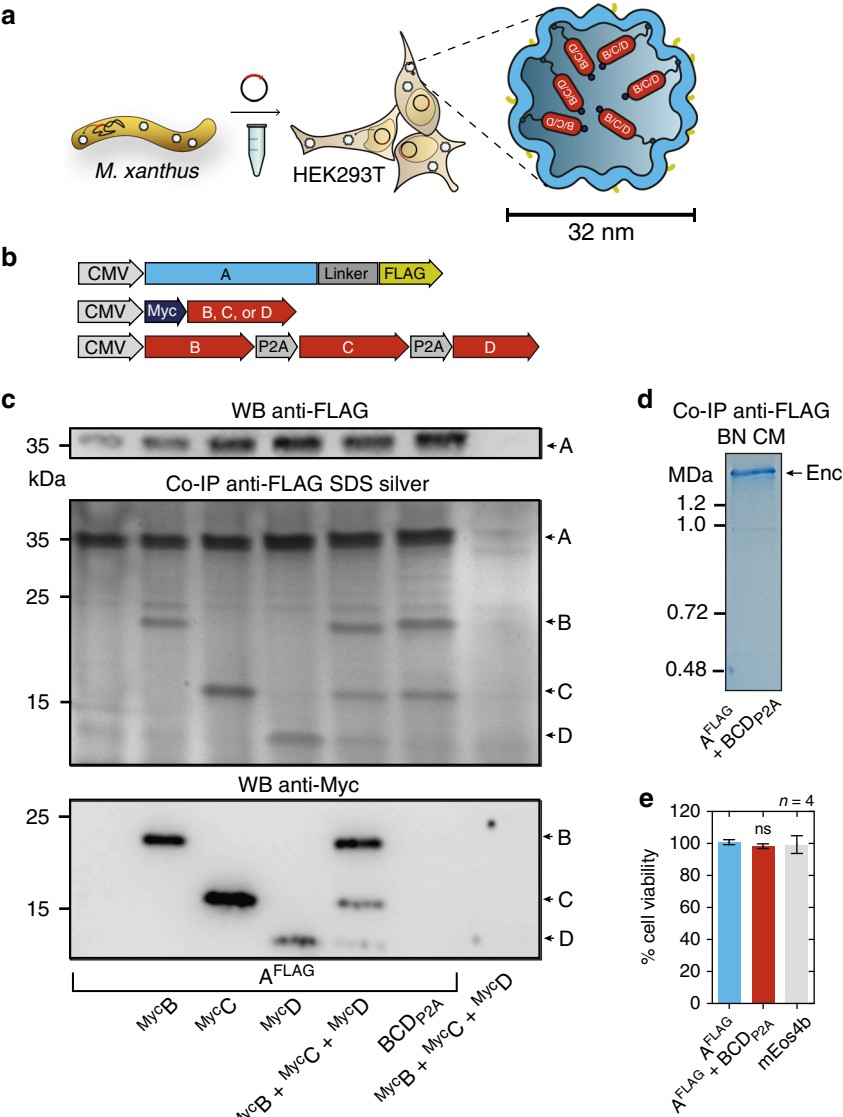

**Fig. 1** Assembly of encapsulins and targeting of cargo in HEK293T cells. **a** Schematic of the heterologous expression of surface-modified encapsulin variants loaded with endogenous cargo proteins. **b** Genetic constructs encoding the shell protein A (light blue) with a FLAG-tag as C-terminal surface modification as well as individual Myc-tagged cargo proteins (red) B, C, and D that can also be combined in a multi-gene expression construct (BCD$_{P2A}$). **c** Co-immunoprecipitation of A$^{FLAG}$ and silver-stained SDS-PAGE from cells co-expressing just B, C, or D, or a combination of these three proteins expressed either via a mixture of individual DNA constructs ($^{Myc}$B + $^{Myc}$C + $^{Myc}$D), or by a multi-gene expression construct (BCD$_{P2A}$). The top panel shows a western blot (WB) against the exterior FLAG-tag in A$^{FLAG}$. The bottom panel shows the corresponding WB against the Myc epitope. **d** Coomassie-stained Blue Native PAGE (BN CM) of purified material from HEK293T expressing A$^{FLAG}$ and BCD$_{P2A}$ yielding a band above 1.2 MDa. **e** Cell viability after 48 h of overexpression of encapsulins (A$^{FLAG}$) with or without cargo (BCD$_{P2A}$) assessed by an LDH release assay. A construct expressing the fluorescent protein mEos4b served as a control. The bars represent the mean ± SEM ($p = 0.1965$, Kruskal–Wallis, $n = 4$; no significant (ns) differences at $\alpha = 0.05$ were found in Dunn's multiple comparisons test between mEos4b and A$^{FLAG}$ expressed without or with BCD$_{P2A}$)

Furthermore, a corresponding Blue Native PAGE (BN-PAGE) of immunoprecipitated FLAG-tagged material from cells expressing A$^{FLAG}$ together with BCD$_{P2A}$ revealed a band with an apparent molecular weight of above 1.2 MDa indicating self-assembly of encapsulin protein complexes and self-targeting of all native cargo proteins (Fig. 1d).

The strong expression of A$^{FLAG}$ without or with loaded cargo did not result in a reduction of cell viability when compared to cells overexpressing a fluorescent protein as assessed by a viability assay based on lactate dehydrogenase (LDH) release (Fig. 1e).

We also generated a construct for a StrepTagII-labeled variant of the shell that co-expresses the ferritin-like Myc-tagged C as cargo protein via a scarless P2A site[41] ($^{Myc}$C-$_{IntP2A}$-A$^{STII}$, Fig. 2a).

Material from HEK293T cells, conveniently purified via Strep-Tactin affinity chromatography, showed assembled nanospheres of 32.4 ± 1.7 nm as the major component in single particle cryo-electron microscopy (cryo-EM) (Fig. 2b, Supplementary Fig. 1a, b), corresponding to the single band >1.2 MDa in size on BN-PAGE (Fig. 2c, right panel). Again, no effect on cell viability was detected for this construct tested by a luciferase-based viability assay compared to A$^{FLAG}$ with and without cargo (BCD$_{P2A}$), as well as controls without expression of encapsulins (EYFP and untransfected HEK293T) (Fig. 2d). Furthermore, N-terminal addition of the human BM40 (osteonectin SPARC) secretory signal peptide (SP) to the StrepTagII-modified encapsulin shell protein resulted in entry into the secretory pathway and robust

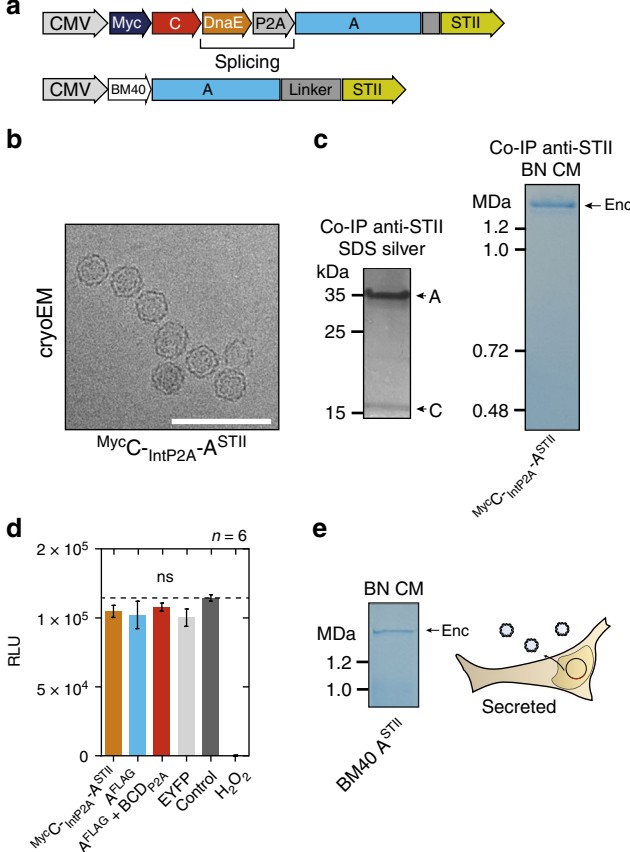

**Fig. 2** Combined encapsulin:cargo construct and secreted encapsulin variant. **a** Scheme of a P2A bicistronic expression construct encoding StrepTagII-tagged ($^{STII}$) nanocompartments containing Myc-tagged C as cargo protein ($^{Myc}$C-$_{IntP2A}$-A$^{STII}$) as well as a variant with an N-terminal BM40 secretion peptide and StrepTagII ($^{STII}$). **b** Cryo-electron microscopy image of material from HEK293T cells expressing $^{Myc}$C-$_{IntP2A}$-A$^{STII}$ purified via Strep-tag II/Strep-Tactin XT affinity chromatography showed the assembled nanospheres of ~32 nm diameter. Scale bar is 100 nm. **c** The corresponding BN-PAGE analysis of the identical material revealed a single band larger than 1.2 MDa. The accompanying silver-stained SDS-PAGE showed the coprecipitation of the cargo $^{Myc}$C with the StrepTagII-modified nanoshell. **d** Luciferase-based cell viability assay after 48 h of overexpression of $^{Myc}$C-$_{IntP2A}$-A$^{STII}$ and A$^{FLAG}$ with or without cargo BCD$_{P2A}$. Cells overexpressing the fluorescent protein EYFP as well as untransfected HEK293T cells served as negative controls. To induce toxicity as positive control, untransfected HEK293T cells were treated with 1 mM $H_2O_2$ 24 h prior to the assay. The bars represent the mean ± SEM ($p$ = 0.442 excluding the positive control, Kruskal–Wallis, $n$ = 6; no significant (ns) differences at $\alpha$ = 0.05 were found in Dunn's multiple comparisons test between any of the encapsulin:cargo conditions and either EYFP or Control). **e** BN CM loaded with cell culture supernatant of HEK293T cells expressing A$^{STII}$ with an N-terminal BM40 secretion signal showed a single band >1.2 MDa

secretion of StrepTagII-modified encapsulins from HEK293T cells as shown by Coomassie-stained BN-PAGE of material present in the cell culture supernatant (Fig. 2e).

**In vivo expression of encapsulins**. To achieve in vivo expression of encapsulins, we generated a coexpression construct that encoded both the nanoshell A$^{FLAG}$ and the ferritin-like protein B from a single plasmid that was small enough to be packaged into an Adeno-associated virus (AAV) (Supplementary Fig. 1c). After

transduction of murine brains via intracranial injections of this viral vector co-expressing A$^{FLAG}$ and B$^{M7}$ by a P2A peptide, we observed robust neuronal expression of the shell protein (Supplementary Fig. 1f, i). Silver-stained BN-PAGE and SDS-PAGE of immunoprecipitated (anti-FLAG) proteins extracted from murine brain showed that the nanocompartments assembled in vivo and that the cargo B$^{M7}$ was associated with the shell (Supplementary Fig. 1g, j). Similar in vivo results could be obtained by co-expressing the nanoshell and ferritin-like B cargo via an IRES site (Supplementary Fig. 1h).

**Encapsulation of engineered cargo**. We next tested whether non-natural cargo molecules could be efficiently targeted into the nanocompartments. We thus C-terminally appended a minimal encapsulation signal, which we found to only necessitate eight amino acids (EncSig), to the photoactivatable fluorescent protein mEos4b[42], coexpressed it with A$^{FLAG}$ and found by co-immunoprecipitation and BN-PAGE analysis under an UV imager that the cargo readily associated with the encapsulin shell (Fig. 3b).

**Selective degradation of non-encapsulated cargo proteins**. Importantly, we could also selectively enrich cargo proteins to the encapsulin lumen by fusing an FKBP12-derived destabilizing domain (DD) that labels the cargo for rapid degradation unless it is shielded from proteasomal machinery[43]. We show that co-expressing A$^{FLAG}$ and DD-mEos4b-EncSig in HEK293T yielded significantly higher mean fluorescence values than DD-mEos4b-EncSig alone, indicating that cargos inside the encapsulin are protected from proteolytic degradation (Fig. 3c, d). Confocal microscopy revealed that coexpression of DD-mEos4b-EncSig with A$^{FLAG}$ shows green fluorescence throughout the cytosol but not in the nucleus, whereas the absence of the encapsulin shell ablated the fluorescence signal. In a positive control in which DD-mEos4b-EncSig was stabilized by adding a small molecule instead of encapsulating it, fluorescence was observed throughout the cell including the nucleus (Fig. 3e).

We then purified encapsulins co-expressed with and without DD-mEos4b-EncSig as cargo to determine their native mass and found that in the absence of cargo, also smaller nanospheres assembled consistent with the known configuration as 60-mers with $T$ = 1 symmetry (Supplementary Fig. 2). We estimated that on average ~60 fluorescent proteins per nanoshell were enclosed as confirmed by gel densitometry (Supplementary Fig. 3a, b). We furthermore found that the FLAG-tagged encapsulin shell is phosphorylated (Supplementary Fig. 3c-e).

**Simultaneous encapsulation of sets of engineered cargo**. We next wanted to assess whether multiple engineered cargo molecules could be encapsulated together. We thus fused the two halves of split PAmCherry1 (PA-s1, PA-s2) to either B or C (B-PA-s2: 27.0 kDa, C-PA-s1: 33.1 kDa) and tested for bimolecular fluorescence complementation (BiFC) within the nanocompartment[44] (Fig. 4a, b). Either of these components could be co-immunoprecipitated with A$^{FLAG}$ as shown by silver-stained SDS-PAGE (Fig. 4b, Supplementary Fig. 4a). The photoactivation of the complemented split PAmCherry1 inside the encapsulins could also be detected via fluorescence imaging of the corresponding BN-PAGE (Fig. 4b, right panel, Supplementary Fig. 4a). Co-expression of both split halves together with A$^{FLAG}$ lead to a strong increase of photoactivatable fluorescent signal throughout the cytosol of HEK293T cells as quantified by confocal microscopy compared to cells that did not express A$^{FLAG}$ (Fig. 4c).

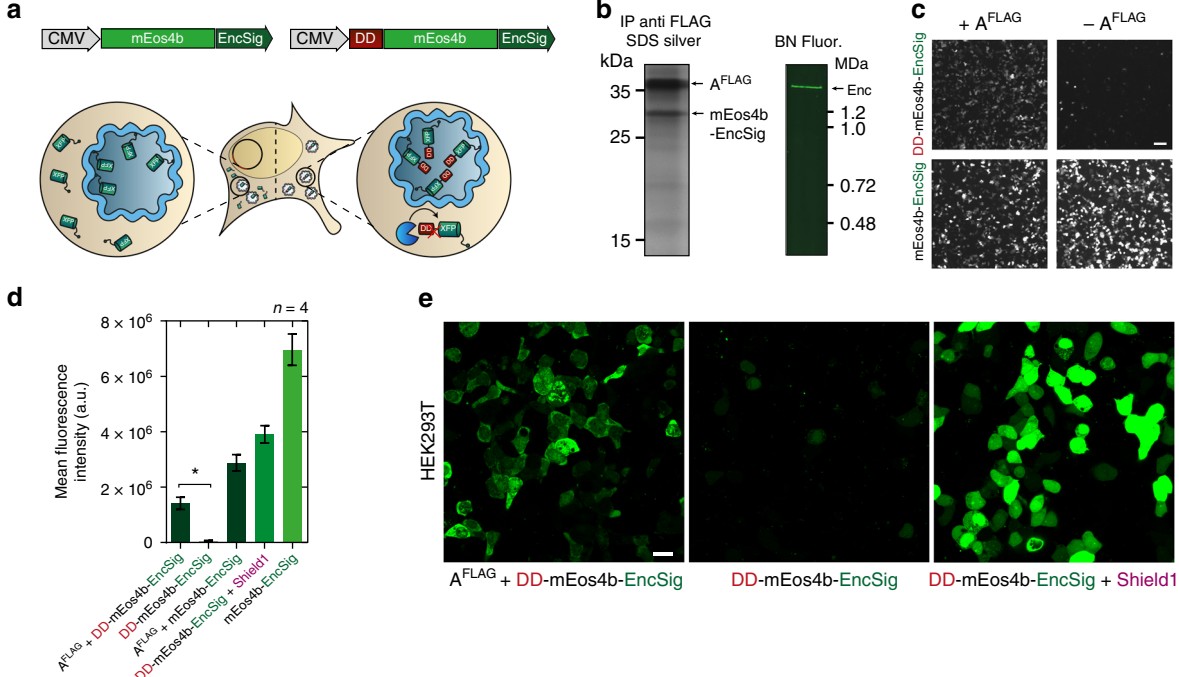

**Fig. 3** Selective degradation of non-encapsulated cargo. **a** Schematic of genetic construct showing a minimal C-terminal encapsulation signal (EncSig) fused to the photoactivatable fluorescent protein mEos4b (mEos4b-EncSig) to associate it to the inner surface of the nanocompartment. When mEos4b-EncSig is N-terminally fused to an FKBP12-derived destabilizing domain (DD), it is degraded by the proteasome unless it is sequestered into the encapsulin shell. This strategy thus selectively enriches cargo inside the lumen of the nanocompartment. **b** Cargo loading of mEos4b-EncSig into the nanocompartment composed of $A^{FLAG}$ was demonstrated by co-immunoprecipitation (Co-IP) against the FLAG epitope followed by silver-stained SDS-PAGE (left panel). Corresponding analysis of whole cell lysate by BN-PAGE on a UV imager shows fluorescence of the native encapsulin band indicating the presence of the mEos4b-EncSig cargo. **c** Representative 2 × 2 table of epifluorescence microscopy images from HEK293T cells co-expressing DD-mEos4b-EncSig with or without $A^{FLAG}$ (upper row) compared to coexpression of the cargo without destabilizing domain (mEos4b-EncSig, lower row). Scale bar represents 50 μm. **d** Corresponding quantification of the fluorescence intensities as exemplified in **c**. Co-expression of $A^{FLAG}$ significantly protects degradation of DD-mEos4b-EncSig ($p = 0.0286$, Mann–Whitney test, $n = 4$ biological replicates, error bars represent mean ± SEM). DD-mEos4b-EncSig can also be stabilized by adding the small molecule Shield1 (0.5 mM, magenta label) to the cell culture medium. **e** Confocal microscopy images of HEK293T expressing DD-mEos4b-EncSig with or without $A^{FLAG}$. As a reference DD-mEos4b-EncSig was stabilized via the addition of 0.5 mM Shield1. Please note that the contrast of all images was linearly adjusted to the same extent optimizing for the condition shown on the left, which resulted in partial oversaturation of the condition shown on the right. Scale bar represents 20 μm

**Compartmentalized enzymatic reactions**. To showcase the use of the eukaryotically expressed encapsulins as bioengineered reaction chambers with pores that can constrain passage of reactants and reaction products, we targeted several enzymes to the nanocapsules. In the presence of $A^{FLAG}$, the split luciferase[45] parts LgBit and SmBit fused to C and B (C-LgBit: 32.7 kDa, B-SmBit: 19.6 kDa) were complemented to functional enzymes as demonstrated by bioluminescence detection from BN-PAGE (Fig. 4d, left) and from total lysate (Fig. 4d, bar graph on the right). Importantly, only very low luminescence signals were detected when the encapsulin shell was not present indicating that using split protein approaches can also ensure confined enzyme activity inside the capsules, in addition to the strategy for selective enrichment of cargo inside the nanocompartment as shown in Fig. 3.

**Bioengineered melanosomes as gene reporters for MSOT**. We subsequently sought to utilize selective passage of small substrates through the nanoshell to load the compartments with tyrosinase as cargo which is the sole enzyme generating the photoabsorbing polymer melanin from the amino acid tyrosine. Because of these attractive features, tyrosinase has been used as a gene reporter for optoacoustic tomography[46,47], an imaging modality that maps the distribution of photoabsorbing molecules in tissue by locating the ultrasonic waves that they emit in response to local heating upon

laser absorption[48,49]. However, melanin production is toxic to cells if not confined in melanosomes, which are membranous compartments of specialized cells[50,51]. We thus chose a soluble tyrosinase from *Bacillus megaterium*[52] that we thought could still be functional as a fusion protein to the native cargo D ($^{Myc}$D-BmTyr: 47.7 kDa) serving as targeting moiety (Fig. 5a). Indeed we could observe generation of melanin on the BN-PAGE band corresponding to the assembled nanocompartment (Fig. 5b). In cells expressing the encapsulin-targeted tyrosinase and the shell $A^{STII}$, we observed robust melanin formation by bright-field microscopy without the strong toxicity apparent in the morphology of control cells expressing just the tyrosinase (Fig. 5c, white arrows). Encapsulation of the tyrosinase also led to a significant increase in cell viability as assessed by a luciferase-based viability assay (Fig. 5d). Cells expressing melanin-producing encapsulins were dark in color (Fig. 5e, inset) and thus generated intense photoacoustic signal even when referenced against strongly absorbing synthetic ink with an optical density of 0.2 (Fig. 5e).

Similarly, we showed that the engineered peroxidase APEX2[53] can polymerize Diaminobenzidine (DAB) when targeted to the nanocompartment (APEX2-EncSig; 31.0 kDa) as indicated by the generation of photoabsorbing DAB polymers associated with the BN-PAGE band corresponding to the assembled nanosphere (Supplementary Fig. 4b).

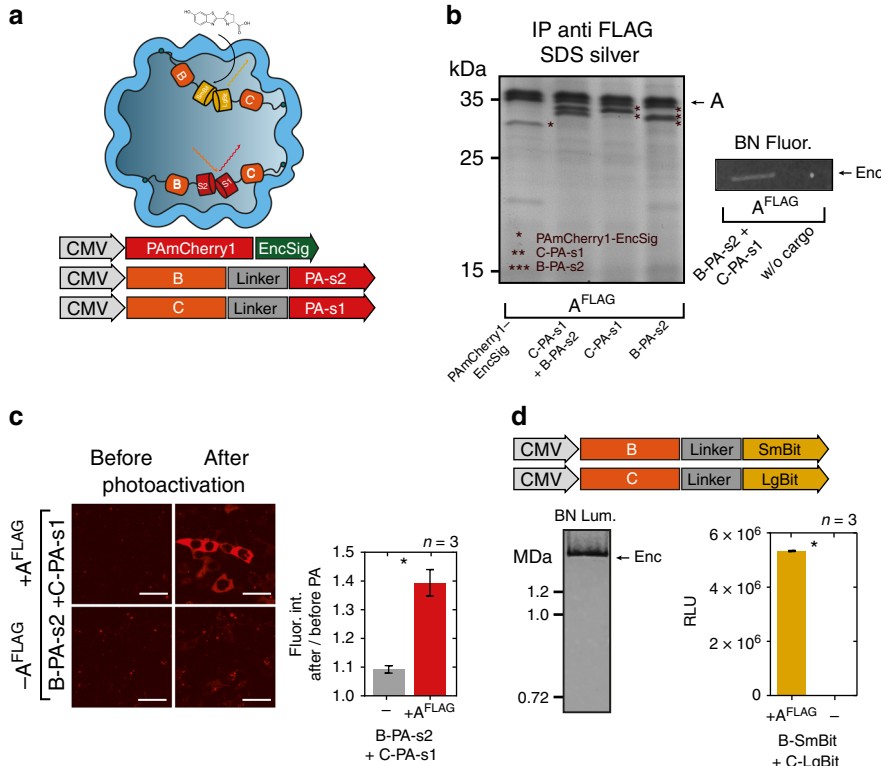

**Fig. 4** Multi-component processes and enzymatic reactions can be targeted to encapsulins in mammalian cells. **a** Overview schematic of sets of cargo molecules for bimolecular fluorescence and enzyme complementation inside the nanocompartment. Targeting of foreign cargo proteins can be achieved either via a minimal C-terminal encapsulation signal (EncSig) or via C-terminal fusions to the native cargo proteins B, C, or D. **b** Silver-stained SDS-PAGE from a co-immunoprecipitation (Co-IP) of A^FLAG co-expressed with photoactivatable mCherry1 with EncSig (PAmCherry1-EncSig) or with either one of the halves of split PAmCherry1 fused to C or B, or a combination of both (C-PA-s1 + B-PA-s2). Fluorescence originating from complemented split PAmCherry1 inside the encapsulins was detected on BN-PAGE loaded with whole cell lysates of cells expressing A^FLAG and C-PA-s1 + B-PA-s2 after 2 min of photoactivation (PA) on an UV imager. **c** Live cell confocal microscopy images (scale bar represents 20 µm) of HEK293T cells expressing B-PAs1 and C-PAs2 with or without the shell-protein A^FLAG before and after 60 s of photoactivation (PA) with 405 nm (upper panel) demonstrating efficient bimolecular fluorescence complementation inside encapsulin compartments. Fluorescence of photoactivated split PAmCherry1 was excited using a 561 nm laser. Fluorescence signals of the sample without and with A^FLAG were quantified by calculating the ratio of the mean signal after PA divided by the signal before PA. The bars in the lower panel represent the mean fluorescence intensity ratios averaged over independent transfection experiments ± SEM ($p = 0.0123$, unpaired $t$-test, $n = 3$). **d** Luminescence signal from BN-PAGE incubated with luciferase substrate and loaded with whole cell lysates of HEK293T co-expressing split luciferase fragments fused to either B or C (B-SmBit, C-LgBit) and A^FLAG (left panel). The luminescent band corresponds to the complemented split luciferase inside the assembled nanocompartment. The bar graph (right panel) shows the corresponding total luminescence signals from the cell lysates expressing B-SmBit and C-LgBit with or without A^FLAG, (mean ± SEM across three independent transfection experiments, in each experiment three technical replicates were averaged, $p < 0.0001$, unpaired $t$-test, $n = 3$)

Since the electrophoretic mobility of protein complexes on BN-PAGE also depends on their hydrodynamic size and shape[54], cargo-loading could be confirmed by observing an identical migration behavior of loaded as compared to unloaded capsules (Supplementary Fig. 4c).

Moreover, we targeted the putative cystathionine γ-lyase (SmCSE) to the nanospheres via an EncSig (smCSE-EncSig: 43.9 kDa) as shown by Co-IP with A^FLAG (Supplementary Fig. 4d). In the presence of L-cysteine, this enzyme was reported to catalyze a conversion of cadmium acetate in aqueous solution into cadmium sulfide (CdS) nanocrystals such that they would generate a photoluminescence signal under UV illumination characteristic for crystal formation at quantum confined sizes[55]. Indeed, we could detect a photoluminescence signal from the BN-PAGE band corresponding to encapsulin loaded with SmCSE-EncSig after on-gel incubation with cadmium acetate and L-cysteine indicating that the smCSE-EncSig cargo was enzymatically active when bound into the shell (Supplementary Fig. 4d).

**Size-constrained iron biomineralization.** Another reason to choose encapsulins from *M. xanthus* was that it was previously reported to deposit iron via the ferritin-like cargo B and C into relatively large compartments (~32 nm, $T = 3$)[31]. We thus investigated whether this functionality could also be realized in eukaryotic cells to enable spatially confined iron deposition sequestered away from the complex signaling network controlling mammalian iron homeostasis. We thus generated a stable cell line co-expressing the nanoshell (A^FLAG) with all native cargo proteins (B,C,D) via a dual-promoter construct (Fig. 6a). In this cell line, we observed long-term and robust expression of all components shown by co-immunoprecipitation with A^FLAG and by immunocytochemistry against the external FLAG-epitope (Fig. 6b, left panel, Supplementary Fig. 5a). Transient co-expression of the ferrous iron transporter MmZip14^FLAG (Zip14) in the stable cell line resulted in a robust dose-dependent iron loading (with ferrous ammonium sulfate (FAS) at concentrations between 0.25–1.25 mM) already after 48 h of supplementation as detected on BN-PAGE via

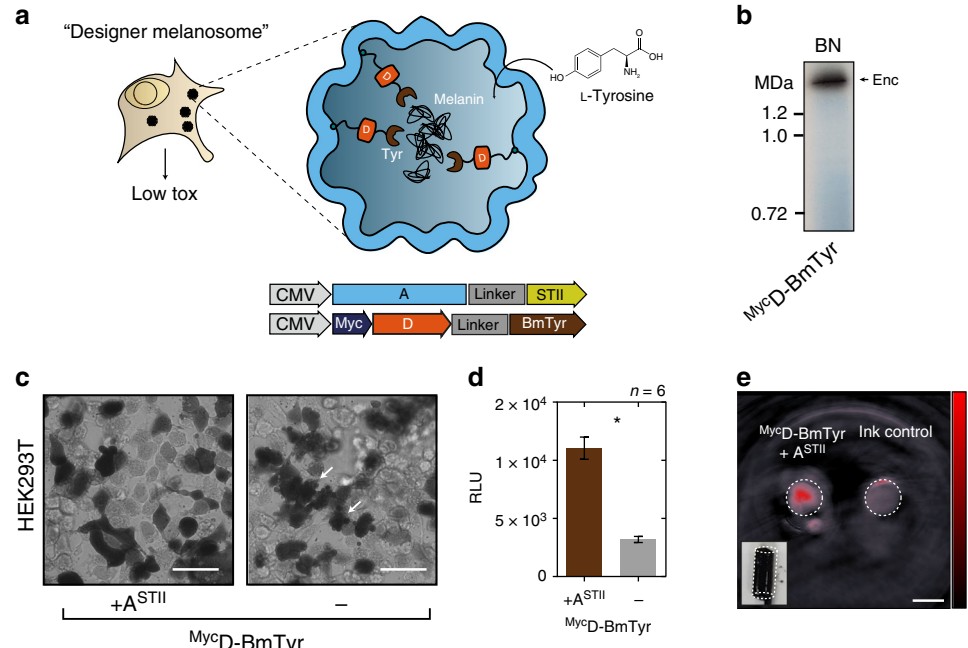

**Fig. 5** Bioengineering of a melanosome by targeting melanin-generating tyrosinase to the encapsulin compartment. **a** Schematic of the detoxifying effects of compartmentalized melanin production by encapsulated tyrosinase from *Bacillus megaterium* targeted to the nanocompartment via fusion to the native cargo D. The substrate ʟ-tyrosine enters the compartment via the pores in the nanoshell. **b** BN-PAGE showing on-gel production of melanin via tyrosinase expressed in HEK293T cells fused to Myc-tagged encapsulin-cargo D (^MycD-BmTyr) to encapsulate it in the assembled nanoshell. Dark colorization of the band was observed after incubation with 2 mM ʟ-tyrosine and 100 μM $CuCl_2$ in PBS (pH 7.4) for 1 h at 37 °C. **c** Bright-field images of HEK293T cells expressing ^MycD-BmTyr with and without StrepTagII-modified shell ($A^{STII}$) after 48 h of expression. Twenty four hours post transfection, cells were supplemented with 1 mM ʟ-tyrosine and 10 μM $CuCl_2$. Cell protrusions (white arrows) were apparent indicating toxic effects of overexpression of non-encapsulated tyrosinase. Scale bar: 20 μm. **d** Corresponding luciferase-based viability assay of HEK293T cells treated as in **c** overexpressing ^MycD-BmTyr with or without $A^{STII}$ after 48 h. (The bars represent the mean ± SEM, $n = 6$, $p < 0.0001$, unpaired $t$-test.) **e** Images of two tubular phantoms (transversal slice) obtained by multispectral optoacoustic tomography (MSOT). The phantoms were filled with ~$10^7$ cells in 1.5% low melting agar expressing ^MycD-BmTyr with $A^{STII}$ (supplementation as in **c** and **d**) or containing highly concentrated ink (OD = 0.2) as control showing the intense contrast obtained between 690 nm and 900 nm from the melanin-producing encapsulins. The coefficients obtained from linear unmixing of the optoacoustic spectra with a melanin reference spectrum are displayed on the red colormap overlaid on the image obtained at 720 nm. The lower left inset shows a color photograph of the tubular phantom containing the cells. Scale bar: 3 mm

DAB-enhanced Prussian Blue staining (DAB PB) (Fig. 6b, right panel).

Efficient iron loading could also be achieved by transient expression of $A^{FLAG}$ + $BCD_{P2A}$ together with Zip14. Under these conditions, iron supplementation with ~0–3 mM FAS for 48 h led to a substantial dose-dependent iron loading of the nanocompartment that saturated at ~1 mM FAS as shown by Coomassie and DAB-enhanced Prussian Blue BN PAGE (Fig. 6c, upper panel, Supplementary Fig. 5b). Interestingly, when we tested the cargo molecules individually for their ability to load iron into the nanosphere, we found that co-expression of only B or C generated equally intense DAB PB bands as compared to $BCD_{P2A}$, indicating that either B or C is sufficient for iron deposition inside the nanocompartment.

In contrast, co-expression of D with $A^{FLAG}$ or any of the cargo molecules without the presence of $A^{FLAG}$ did not lead to discernable DAB PB signals (Fig. 6c, lower panel, Supplementary Fig. 5c). In a standard cell viability assay, we found no impairment of the cells when Zip14 was co-expressed together with $A^{FLAG}$ and $BCD_{P2A}$ or just B. However, ~7% of cells showed reduced viability when the cargos $BCD_{P2A}$ were expressed without the nanocompartment ($p = 0.0238$, Mann Whitney test, $n = 3$) or when only the fluorescent protein mEos4b-EncSig was expressed (Supplementary Fig. 5d) indicating that in the absence of the nancompartment the imported iron was not sufficiently sequestered by the endogenous iron homeostasis machinery.

We furthermore tested variants of A with N-terminal fusions with peptide sequences from *Magnetospirillum magneticum* Mms (6 and 7) proteins reported to aid in templating iron mineralization[56] but found no additional benefit of these modified inner surfaces over $A^{FLAG}$ using our current readout (Supplementary Fig. 5e). In addition, we analyzed several variants of the cargo proteins B and C, fused C-terminally to peptides from Mms proteins (superscripts M6, M7, please see Supplementary Fig. 5f). These data confirmed that either B or C are sufficient to load the nanocompartment with iron and showed that no obvious additional iron loading resulted from the presence of the Mms peptides.

**Encapsulins enable detection via MRI and magnetic sorting.** Next, we were interested in whether the strong iron accumulation inside eukaryotically expressed encapsulin shells would yield significant contrast by MRI. We thus expressed $A^{FLAG}$ alone or together with either all native cargos $BCD_{P2A}$ or just ^MycB, or ^MycD and Zip14 and subjected cell pellets to relaxometry measurements by MRI. The nanocompartment $A^{FLAG}$ co-expressed with all native cargo proteins (BCD) lead to a significant increase in $R_2^*$-relaxation rates as compared to just $A^{FLAG}$. The same effect was observed by co-expressing just the ferritin-like B (Fig. 7a, $p = 0.0047$, Kruskal–Wallis with significant differences at α = 0.05 from Dunn's multiple comparisons test vs. $A^{FLAG}$,

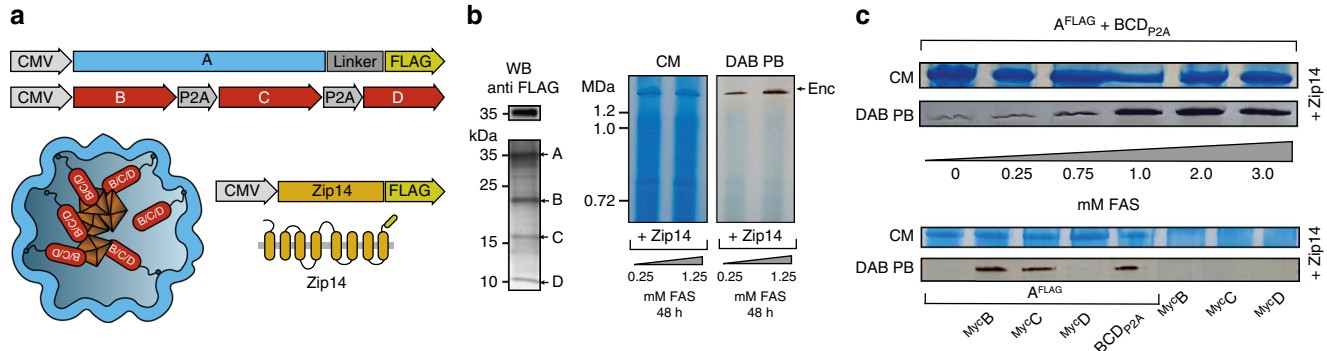

**Fig. 6** Efficient iron loading of eukaryotically expressed encapsulin nanospheres **a** Schematic of a dual-promoter construct used for generation of a stable cell line expressing $A^{FLAG}$ and all native cargos B, C, and D. Also depicted is a construct encoding the iron-transporter MmZip14$^{FLAG}$ (Zip14) used to transport additional amounts of iron into the cell. **b** Co-immunoprecipitation (Co-IP) against the FLAG epitope from a whole cell lysate of a stable HEK293T clone expressing $A^{FLAG}$ together with B, C, and D analyzed by silver-stained SDS PAGE and the corresponding WB against the FLAG epitope (left panel). The pair of Blue Native (BN) gels visualizes proteins from whole cell lysates via Coomassie staining (CM) (left panel) and iron content via treatment with DAB enhanced Prussian Blue (DAB PB) (right panel) from the same stable cell line. Robust iron loading of the assembled nanocompartments was achieved by transient co-expression of MmZip14$^{FLAG}$-$_{IRES}$-ZsGreen1 in which case 0.25 mM ferrous ammonium sulfate (FAS) for 48 h was sufficient to see strong iron loading. **c** BN gel stained with CM or DAB PB loaded with whole cell lysates of HEK293T cells transiently expressing $A^{FLAG}$ + BCD$_{P2A}$ and Zip14$^{FLAG}$ supplemented with different concentrations of FAS (0–3 mM) for 48 h (upper panel). The strong bands, which correspond to the assembled nanoshell, indicate high expression levels of encapsulins and efficient, dose-dependent iron loading. The lower panel shows a CM and DAB PB-stained BN gel from whole cell lysates of HEK293T cells expressing Zip14$^{FLAG}$ and different combinations of native cargo molecules: $^{Myc}$B, $^{Myc}$C, and $^{Myc}$D alone, or all three (BCD$_{P2A}$) with or without $A^{FLAG}$. The robust DAB PB stains show that the ferritin-like cargo proteins B or C are sufficient for iron loading into encapsulins. FAS was supplemented at 2.5 mM for 48 h

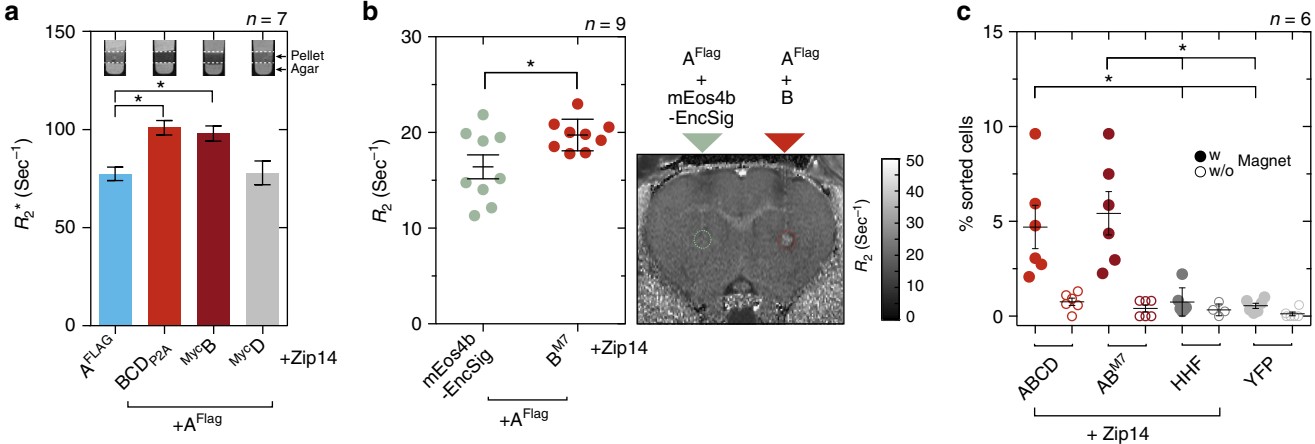

**Fig. 7** Iron-filled encapsulins enable detection by MRI and magnetic cell separation. **a** Relaxometry measurements by MRI conducted on cell pellets (~10$^7$ cells) from HEK293T cells transiently expressing $A^{FLAG}$ + BCD$_{P2A}$, $^{Myc}$B, or $^{Myc}$D, or $A^{FLAG}$ alone (1 mM FAS for 24 h and expression of Zip14$^{FLAG}$). Expression of $A^{FLAG}$ with BCD$_{P2A}$ or with $^{Myc}$B showed a significantly enhanced $R_2^*$-relaxation rate as compared with $A^{FLAG}$ alone or loaded with $^{Myc}$D; ferritin-like cargo B was sufficient to generate an increase in $R_2^*$ in the presence of the $A^{FLAG}$ nanocompartment ($p = 0.0047$, Kruskal–Wallis, $n = 7$ from four independent experiments, stars indicate significance at $\alpha = 0.05$ from Dunn's multiple comparisons test vs. $A^{FLAG}$; the bars represent the mean ± SEM). The insets show MRI slices (13.5 ms echo time) through test tubes in which cells were pelleted on a layer of agar. **b** In vivo MRI detection of HEK293T cells transiently co-expressing $A^{FLAG}$ together with ferritin-like $B^{M7}$ that were xenografted into rat brains. As compared to cells co-expressing $A^{FLAG}$ together with the fluorescent protein mEos4b-EncSig as control cargo, we observed significantly increased transverse relaxation rates ($p = 0.0078$, Wilcoxon matched-pairs signed rank test, $n = 9$) measured at the injection site for $A^{FLAG}$ + $B^{M7}$ expressing cells 24 h post injection. The horizontal lines represent the mean ± SEM. The image on the right shows a coronal $R_2$ map through a rat brain with the regions of interest (ROIs) defined over the injection sites by dashed circles. **c** HEK293T cells were co-expressing $A^{FLAG}$ and BCD$_{P2A}$, $A^{FLAG}$ and $B^{M7}$, or human H-chain ferritin (HHF) as a control together with Zip14 and were treated with 2.5 mM FAS for 48 h. Additional control cells were expressing only EYFP. Independent cell suspensions were subsequently sorted on commercial magnetic separation columns inside and outside the magnetic field to control for unspecific retention in the mesh of the column. The fraction of cells separated in the magnetic field for both encapsulin:cargo conditions was significantly higher than for any of the control conditions HHF + Zip14 or YFP ($p = 0.0007$, Kruskal–Wallis, $n = 6$ from three independent experiments, stars indicate significance at $\alpha = 0.05$ from Dunn's multiple comparisons test across all conditions with magnetic field; the horizontal lines represent the mean ± SEM)

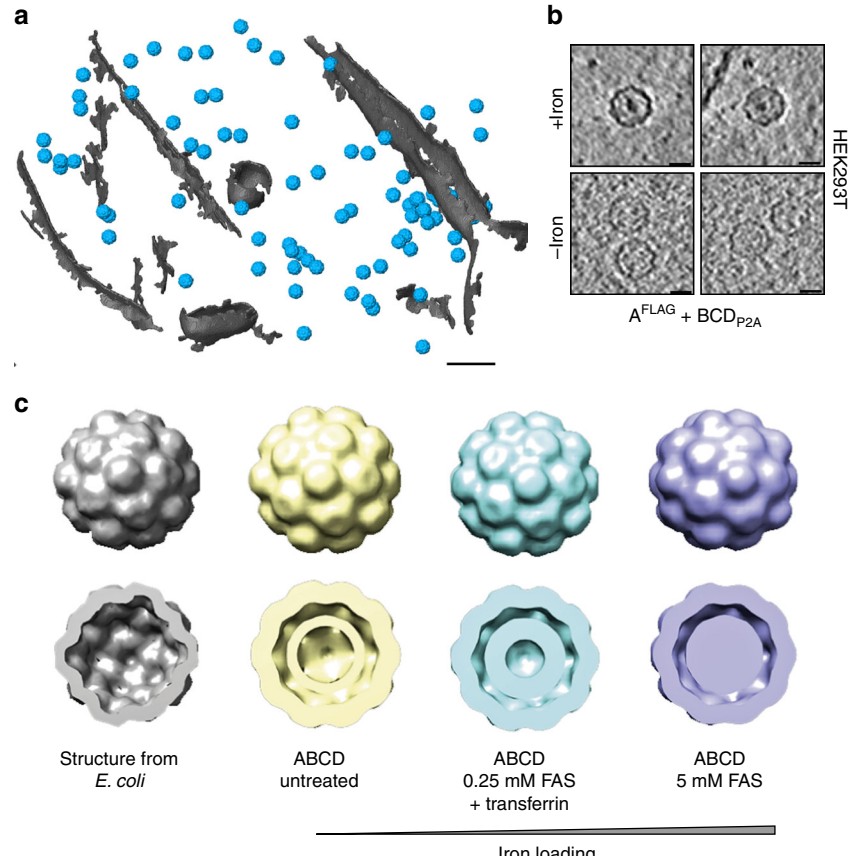

**Fig. 8** Encapsulins as genetically encoded markers for cryo-electron tomography (Cryo-ET). **a** Cryo-ET data from HEK293T cells stably expressing encapsulins together with native ferritin-like cargo proteins (using the dual promoter construct $A^{FLAG}$;$BCD^{P2A}$ shown in Fig. 6a). 3D rendering showing encapsulins in blue and membranes in gray colors. Scale bar:100 nm. **b** Example slices from tomograms show encapsulins with and without electron-dense cores from iron-accumulation (treatment with 5 mM FAS for 48 h prior to vitrification). Scale bars:20 nm. **c** In situ structures (displayed as 2× binned) derived from cryoelectron tomography of nanocompartments assembled in HEK293T cells without (beige), with 0.25 mM FAS and 1 mg/ml human transferrin (cyan) and 5 mM FAS (purple) as compared with the published structure shown in gray (pdb 4PT2; EMDataBank EMD-5917) that was obtained from *M. xanthus* EncA expressed in *E. coli*[31]. The cutaway views of the encapsulins show electron densities indicating the presence of cargo proteins (beige) and additional iron deposition (cyan and purple) as compared to published data from the *EncA* shell[31] that were obtained in the absence of cargo proteins

$n = 7$). This indicated again that co-expression of B was sufficient to generate efficient iron deposition inside the nanoshell.

We subsequently sought to test whether cells genetically labeled with encapsulins could be detected by MRI in vivo. As an initial assessment, we thus xenografted cells co-expressing $A^{FLAG}$ together with $B^{M7}$ into rat brains and obtained $R_2$-relaxation maps that showed elevated relaxation rates ($p = 0.0078$, Wilcoxon matched-pairs signed rank test, $n = 9$) at the injection site as compared to xenografted cells in which the fluorescent protein mEos4b-EncSig was used as a control cargo (Fig. 7b).

In addition to MRI contrast, the iron biomineralization inside the encapsulins also allowed us to magnetically sort cells co-expressing the shell $A^{FLAG}$ with $BCD_{P2A}$ or with $B^{M7}$ at significantly higher percentages than when human H-chain ferritin (HHF) was expressed or just yellow fluorescent protein (EYFP) ($p = 0.0007$, Kruskal–Wallis, with significant differences at $\alpha = 0.05$ from Dunn's multiple comparisons test, $n = 6$; Fig. 7c).

**Encapsulins as markers for electron microscopy.** Given that the iron loading of the encapsulins was very efficient and observable at the population level, we next assessed how well individual nanocompartments could be detected by electron microscopy in cells such that they could be used as genetically encoded markers. We thus grew HEK293T cells stably expressing the shell protein

$A^{FLAG}$ and $BCD_{P2A}$ using a dual promoter vector on a transmission electron microscopy (TEM) grid, vitrified them by plunge-freezing and produced lamellae by cryo-focused ion beam (cryo-FIB) milling for in situ cellular cryo-electron tomography (cryo-ET). The heterologously expressed encapsulins were readily detected as clearly discernible nanospheres (Fig. 8a, Supplementary Fig. 6a, b) that exhibited electron-dense cores when we supplemented the growth media with ferrous iron (Fig. 8b) and were distributed as monodisperse spheres throughout the cytosol (Supplementary Fig. 6c, d). The electron density maps showed a high similarity to the structure published from encapsulin shells from *M. xanthus* expressed in *E. coli* (pdb 4PT2; EMDataBank EMD-591728, Fig. 8c). The clipped views from the encapsulins (blue) furthermore show electron densities associated with docked cargo proteins and most likely biomineralized iron as compared with the inner surface of the shell from *E. coli* (gray) that was mapped in the absence of any cargo (Fig. 8c, lower row). These data demonstrate that the spherical shape and high, non-toxic expression levels make encapsulin very attractive as fully genetically expressed markers for EM.

## Discussion

In summary, we genetically controlled multifunctional orthogonal compartments in mammalian cells via expressing N- or

C-terminally modified encapsulins, which we found to auto-assemble into abundant nanocompartments which readily encapsulated sets of natural and engineered cargo proteins and enabled size-constrained metal biomineralization.

The efficiency of self-targeting and auto-packaging of the various cargo proteins in mammalian cells was remarkable given that the number of possible protein interactions is even a few-fold higher than in the original prokaryotic host organism[57,58]. We found that about 60 cargo proteins of a canonical fluorescent protein can be bound to the inner encapsulin surface via the minimal encapsulation signal. Higher loading factors could be achieved by providing cargo proteins with multidentate adapters such that the entire encapsulin volume could be filled. We furthermore observed that without co-expression of endogenous or engineered cargo, the abundance of the 60-mer encapsulin shell with $T = 1$ symmetry was increased, which is in line with a previous observation made from *M. xanthus* encapsulin expressed in *E. coli*[31] and suggests that encapsulation of cargo leads to the preferred assembly of the 180-mer in $T = 3$ symmetry.

Also, iron storage inside the capsule via the ferritin-like enzymes B or C targeted to the encapsulins was very efficient, indicating that there was sufficient access to ferrous iron. Whereas encapsulins heterologously expressed in *E. coli* were shown to load iron[59], this could so far not be shown in mammalian cells. We also found that just co-expression of B (or C) with A is sufficient for robust iron storage such that a single-piece reporter construct of just ~2.1 kb in size can be used.

In the context of optimizing $T_2$ contrast in MRI, it would certainly be valuable to explore modifications of the outer surface that may control the agglomeration state and thus could modulate the apparent relaxivity of encapsulin ensembles[60]. In this context, it would also be desirable to explore capsid architectures with more storage capacity such as ones with $T = 7$ quasisymmetry known from bacteriophage HK97[31,61]. Furthermore, modifications of the inner surface of the shell may be engineered and/or additional cargo could be designed that could facilitate the nucleation process to support higher iron packing densities or alter environmental parameters (e.g., pH and redox potential) to potentially even generate superparamagnetic iron-oxides which possess a substantially larger magnetization[62]. Iterative optimization schemes such as directed evolution could also be employed based on rescue assays from excess iron or magnetic microfluidic sorting and could also be complemented by parallel screens in prokaryotes if enough iron-influx can be achieved there.

Their dense monodisperse distribution, spherical shape, and sufficient size, also render encapsulins excellent genetically expressed EM markers in mammalian cells (Supplementary Movie 1) that are much more readily detectable than ferritins, which have been visualized by EM in *E. coli* and yeast[63,64]. In addition, the iron-based contrast in encapsulins has the advantage over semi-genetic methods such as metallothionein (MT), mini-iSOG, erHRP, or APEX/APEX2 that no fixation and delivery of artificial substrates and precipitation of electron-dense material is necessary which may alter cellular structures[53,65–68]. Instead, if iron-based EM contrast is desired, cells expressing the iron-accumulating encapsulins can just be grown in regular growth media containing sufficient iron for transferrin-mediated uptake before direct plunge freezing and cryo-EM.

For future applications as EM gene reporters in, e.g., connectomics research, it would be desirable to generate further encapsulin variants with surface-presented targeting moieties to control their subcellular localization. In this regard, it is of note that virtue of the self-assembling mechanism, the size of A is only 0.9 kb and that of B just 0.5 kb such that a combined construct is small enough to be carried by viruses optimized for trans-synaptic tracing[69]. It should furthermore be feasible to perform selective detection of encapsulins loaded with split photo-activatable fluorescent proteins via photoactivated localization microscopy (PALM) and combine this with cryo-ET as was demonstrated for photoactivatable GFP (cryo-PALM)[70].

Besides allowing the influx of metals for size-constrained biomineralization for the type of applications discussed above, the pore size of ~5 Å inside the encapsulin shell also affords selective passage of small substrates, whereas reaction products may be trapped inside the nanoshell. We have exploited this feature by encapsulating tyrosinase for confined enzymatic production of the toxic polymer melanin and utilized the engineered "nano-melanosomes" as genetically encoded reporters for optoacoustic imaging.

In future applications, encapsulins could thus be used as versatile reaction chambers for, e.g., metabolic engineering of orthogonal reactions in eukaryotic cells. The toolbox for genetically controlled compartmentalization in mammalian cells which we introduce here could, for instance, enable multi-step enzymatic production involving labile or toxic intermediates but yielding end-products that may have beneficial intracellular effects or serve as molecular signals upon "quantal" release from the nanocompartment. The approach could for instance also endow genetically modified mammalian cells used for cell therapies with metabolic pathways that may augment their therapeutic efficacy. Complementarily, endogenously produced toxic products could be contained and detoxified in engineered compartments for causal studies or potentially for cell or gene therapies.

In addition to the encapsulins presented here, heterologous expression of compartments with different sizes and shapes seem possible, which could offer different sets of endogenous and engineered cargo molecules with different subcellular targeting. These alternative systems would ideally also be orthogonal to each other such that multiplexing (maybe even nesting) of several engineered compartments and multicomponent processes could be achieved.

More generally, genetically controlled compartmentalization of multi-component processes in eukaryotic cells—as demonstrated for encapsulins here—is a fundamental biotechnological capability that has profound implications for mammalian cell engineering and emerging cell therapies.

## Methods

**Genetic constructs.** Mammalian codon-optimized MxEncA (UniProt: MXAN_3556) MxEncB, MxEncC, and MxEncD (UniProt: MXAN_3557, MXAN_4464, MXAN_2410) were custom synthesized by Integrated DNA Technologies and cloned into pcDNA 3.1 (+) Zeocin (Invitrogen) using restriction cloning or Gibson assembly. The MxEncA surface tags (FLAG or StrepTagII) were C-terminally appended using Q5® Site-Directed Mutagenesis (New England Biolabs). N-terminal Myc epitopes were added accordingly to the cargo proteins. Multigene expression of B, C, and D was achieved by generating a single reading frame containing all three genes separated by P2A peptides yielding BCD$_{P2A}$. A "scarless" bicistronic construct encoding $^{Myc}$C and A$^{STII}$ was custom synthesized by inserting a *Ssp* DnaE mini-intein variant engineered for hyper-N-terminal autocleavage followed by a P2A peptide in between the genes as previously described[41]. For generating stable clones expressing MxEncABCD, MxEncA$^{FLAG}$ was cloned into the Cytomegalovirus promoter (CMV) driven expression cassette of pBudCE4.1 (Invitrogen) and BCD$_{P2A}$ was cloned into the elongation factor 1 alpha promoter (EF1a) driven expression cassette of the vector via restriction cloning. To generate AAV enabling multigene expression of MxEncA$^{FLAG}$ and MxB-Mms7ct, two strategies were employed: MxEncA$^{FLAG}$ was cloned upstream of an ECMV internal ribosome entry site (IRES) whereas MxB-Mms7ct was inserted downstream. The second approach employs MxB-Mms7ct followed by a P2A peptide and MxEncA$^{FLAG}$. The two cassettes were subcloned into pAAV-CamKIIa (https://www.addgene.org/26969/) with BamHI and EcoRI. AAVs were custom prepared by the UNC Vector Core of the University of North Carolina at Chapel Hill. To test the bicistronic expression constructs used for the AAVs in HEK293T cells, the cassettes were also sub-cloned into the pcDNA 3.1 (+) Zeocin with EcoRI and NotI. To target PAmCherry1 and mEos4b as cargo to the encapsulin nanocompartments, the fluorescent proteins were C-terminally fused to 2 × GGGGS linkers followed by the minimal encapsulation signal LTVGSLRR

(EncSig). To generate the destabilized version of mEos4b, the L106P mutant of FKBP12 (DD-N)[43] was N-terminally appended to mEos4b-EncSig using Gibson-Assembly yielding DD-mEos4b-EncSig. For complementation of split PAm-Cherry1 inside the encapsulin nanoshell, amino acids 1–159 of PAmCherry1 were fused to MxEncC via a $2 \times$ GGGGS linker and amino acids 160–236 of PAm-Cherry1 were directly fused to the C-terminus of MxEncB. For complementation of a split luciferase, the split part LgBit (NanoBiT system, Promega) was fused C-terminally to MxEncC via a $2 \times$ GGGGS linker. SmBit was directly fused to the C-terminus of MxEncB. SmCSE[55] (UniProt: Smal_0489) and APEX2[53] were fused to $2 \times$ GGGGS linker followed by the minimal encapsulation signal. Mammalian codon-optimized *Bacillus megaterium* tyrosinase (BmTyr) was C-terminally appended to $^{Myc}D$ separated by $2 \times$ GGGGS linker in custom gene synthesis. C-terminally FLAG-tagged *Mus musculus* Zip14 was inserted into pcDNA 3.1 ( + ) or pIRES2-ZsGreen1 via restriction cloning. To yield secreted encapsulins, MxEncA$^{STII}$ was N-terminally fused to a human BM40 secretion peptide. In order to generate encapsulin derivatives featuring C-terminal acidic peptides of magnetotactic bacteria Mms proteins that are implicated in mediation of magnetite formation either the C-terminal peptide of Mms6 (YAYMKSRDIESAQSDEEVELRDALA) or Mms7 (YVWARRRHGTPDLSDDALLAAAGEE) of *Magnetospirillum magneticum* were fused either to the inward-facing N-terminus of MxEncA$^{FLAG}$ or to the C-terminus of either the MxEncB or C using Q5® Site-Directed Mutagenesis. For a complete list of the genetic constructs featuring their composition refer to Supplementary Table 1.

**Cell culture.** Low passage number HEK293T (ECACC: 12022001, obtained via Sigma-Aldrich) and CHO (ECACC: 85050302, obtained via Sigma-Aldrich) cells were cultured in advanced DMEM with 10 % FBS and penicillin–streptomycin at 100 μg/ml at 37 °C and 5% CO$_2$. Cells were transfected with X-tremeGENE HP (Roche) according to the protocol of the manufacturer. DNA amounts (ratio shell to cargos) were kept constant in all transient experiments to yield reproducible DNA-Lipoplex formation. To generate a stable HEK293T cell line expressing MxEncABCD, cells were transfected with pBudCE4.1 MxEncABCD and stable transfectants were selected with 300 μg/ml Zeocin (InvivoGen).

**Protein expression and lysis.** Cells were harvested between 24 and 48 h post transfection. Cells were lysed with M-PER Mammalian Protein Extraction Reagent (Pierce Biotechnology) containing a mammalian protease inhibitor cocktail (SIGMA P8340, Sigma-Aldrich) according to the protocol of the manufacturer in all experiments using FLAG-tagged encapsulins. For lysis of cells expressing StrepTagII-modified encapsulins, cells were resuspended in Buffer W (150 mM NaCl, 100 mM Tris-Cl, pH 8.0) and exposed to four freeze–thaw cycles in LN$_2$. After spinning down cell debris at $10,000 \times g$ for 15 min, cell lysates were kept at 4 °C for downstream analyses. Protein concentrations of lysates were determined by measuring OD at 280 nm.

**Co-immunoprecipitation of encapsulins.** Cell lysates were incubated with Anti-FLAG® M2 Magnetic Beads or Anti-FLAG® M2 affinity gel (SIGMA M8823 and A2220, Sigma-Aldrich) according to the protocol of the manufacturer. After binding, the magnetic beads were washed four times on a magnetic separator rack (DYNAL separator, Invitrogen) with M-PER buffer. Bound FLAG-tagged encapsulins were eluted using M-PER buffer containing 100 μg/ml FLAG-peptide (SIGMA F3290, Sigma-Aldrich). In the case of encapsulins with an external StrepTagII, MagStrep "type3" XT beads or Strep-Tactin®XT resin (IBA Life-sciences) was used according to the protocol of the manufacturer. Proteins were eluted using Buffer BXT (150 mM NaCl, 100 mM Tris-Cl, pH 8.0, 50 mM Biotin). To yield the eluted proteins, samples were mixed with SDS-PAGE sample buffer and incubated at 95 °C for 5 min. Samples were loaded onto pre-cast 12% Bio-Rad Mini-PROTEAN® TGX™ (Bio-Rad Laboratories) gels and run for 45 min at 200 V. Accordingly, gels were either directly silver-stained using SilverQuest™ Silver Staining Kit (Novex) according to the protocol of the manufacturer or immuno-blotted onto PVDF membranes. After blotting, membranes were blocked in 5% non-fat milk in TBS for 1 h at room temperature. Subsequently, membranes were incubated in TBS containing 5% non-fat milk and 1 μg/ml Monoclonal ANTI-FLAG® M2 antibody (SIGMA F1804, Sigma-Aldrich) or 1 μg/ml Anti-Myc Tag Antibody clone 9E10 (05–419, EMD Millipore) for 2 h at room temperature. After five washing cycles with TBS, membranes were incubated with anti-mouse IgG HRP-conjugate (SIGMA A5278, Sigma-Aldrich) for 1 h at room temperature in 5% non-fat milk in TBS. Protein bands were detected using Amersham ECL Prime Western Blotting Detection Reagent (GE Healthcare Bio-Sciences AB) on a Fusion FX7/SL advance imaging system (Peqlab Biotechnologie GmbH). For dephosphorylation of protein material from the Co-IP, 10 units of calf intestinal phosphatase (New England Biolabs) were added to protein solutions in 1× CutSmart Buffer (New England Biolabs) and incubated for 1 h at 37 °C. For densitometric determination of SDS-PAGE bands, band intensity integrals were measured using ImageJ (NIH).

**Blue Native gel electrophoresis and on-gel analyses.** For detection of native encapsulin nanocompartments, the NativePAGE™ Novex® Bis-Tris Gel System (Life Technologies) was used. Either eluted material from the Co-IP/purification or whole cell lysates of cells expressing encapsulins in NativePAGE™ Novex® sample buffer were loaded onto pre-cast NativePAGE™ Novex® 3–12% Bis-Tris gels. NativeMark™ Unstained Protein Standard (Life Technologies) covering a size range between 20 and 1200 kDa was used as a marker. The total protein amount of whole cell lysates loaded per well was adjusted to ~1–3 μg. Blue native (BN) gels were run for 90–180 min at 150 V according to the protocol of the manufacturer. Gels loaded with samples from Co-IP/purification were silver-stained using SilverQuest™ Silver Staining Kit (Novex) or Coomassie-stained using Bio-Safe™ Coomassie Stain (Bio-Rad Laboratories). For protein detection, gels loaded with whole cell lysate samples were Coomassie-stained accordingly. For detection of iron-containing proteins, gels loaded with samples containing iron loaded encapsulins were Prussian Blue (PB) stained. Briefly, gels were incubated in 2% potassium hexacyanoferrate(II) in 10% HCl for 45 min. For 3,3′-diaminobenzidine-enhancement (DAB PB), gels were washed three times with ddH$_2$O and incubated in 0.1 M phosphate buffer (pH 7.4) containing 0.025% DAB and 0.005% H$_2$O$_2$ until dark-brown bands appeared. To stop DAB polymerization, gels were washed three times with ddH$_2$O. For detection of fluorescent signals from native encapsulin bands (fluorescent cargos: mEos4b, PAmCherry1, split PAmCherry1 or mineralized CdS), unstained BN gels were imaged on a Fusion FX7/SL advance imaging system (Peqlab Biotechnologie GmbH) using the UV fluorescence mode. For on-gel detection of luminescence signal generated by encapsulated split NanoLuciferase, unstained BN gels were soaked in 1 ml of Nano-Glo® Luciferase substrate (Nano-Glo® Luciferase Assay, Promega) and imaged on a Fusion FX7/SL advance imaging system (Peqlab Bio-technologie GmbH) in chemiluminescence mode. For whole cell lysate lumine-scence detection, cell lysates were mixed with the substrate at a 1:1 ratio and luminescence readings were taken on a Centro LB 960 (Berthold Technologies) at 0.1 s acquisition time. For detection of APEX2 peroxidase activity inside encap-sulins, unstained BN gels were incubated in 0.1 M phosphate buffer (pH 7.4) containing 0.025% DAB and 0.005% H$_2$O$_2$ for 15 min until black bands appeared on the gel. For microscopic detection of DAB polymerization in cells expressing APEX2-loaded encapsulins, cells were fixed in 4% PFA in PBS for 15 min. Sub-sequently, cells were incubated in 0.1 M phosphate buffer (pH 7.4) containing 0.025% DAB and 0.005% H$_2$O$_2$ for 5 min. The reaction was stopped by washing three times with PBS. For the on-gel detection of melanin generation associated with encapsulins, gels loaded with whole cell lysates of HEK293T cells expressing encapsulins loaded with tyrosinase were incubated in PBS containing 2 mM L-tyrosinase and 100 μM CuCl$_2$ for 1 h at 37 °C until a black encapsulin band became visible.

**Size exclusion chromatography.** Size exclusion chromatography (SEC) of purified A$^{FLAG}$ with or without DD-mEos4b-EncSig was performed on an Äkta Purifier (GE Healthcare) equipped with an analytical size exclusion column (Superose 6 10/300 GL, GE Healthcare) at 4 °C. For refractive index (RI) detection, a Viscotek TDA 305 triple array detector (Malvern Instruments) downstream of the column was used. In total, 100 μl samples were run at a flow rate of 0.4 ml/min in 50 mM Tris-HCl, 150 mM NaCl, 1 mM EDTA, pH 7.4, at a concentration of 0.3 mg/ml.

**Dynamic light scattering.** Dynamic light scattering experiments were performed on a DynaPro NanoStar instrument and analyzed with DYNAMICS 7.1.9 software (Wyatt Technology). Measurements were performed at 22 °C using standard rec-tangular cuvettes containing 60 μl of protein sample in the concentration range between 0.15 and 0.5 mg/ml. For each measurement, 100 acquisitions with an acquisition time of 5 s were recorded.

**Native mass spectrometry.** Purified sample material from HEK293T cells expressing A$^{FLAG}$, with and without co-expression of the photoactivatable fluor-escent protein DD-mEos4b-EncSig, was buffer exchanged to 150 mM aqueous ammonium acetate, pH 7.5 using Micro Bio-Spin Columns with Bio-Gel P6 (Biorad, USA) following the manufacturer's protocol for buffer exchange. Samples were analyzed at a concentration of 0.1–0.45 g/l, corresponding to an estimated monomer concentration ranging from 3 to 14 μM. Gold-coated nanoelectrospray needles were made in-house from borosilicate capillaries (Kwik-Fil, World Preci-sion Instruments, Sarasota, FL) on a P97 puller (Sutter Instruments, Novato, CA) and being coated by using an Edwards Scancoat six pirani 501 sputter coater (Edwards Laboratories, Milpitas, USA). Measurements were carried out in positive ion mode on a modified Q-ToF 2 (Waters, UK) instrument[71,72], operated at ele-vated pressure in the source region (~10 mbar), using Xenon as collision gas at $2*10^{-2}$ mbar in the collision cell. Capillary and sample cone voltage was set to 1400 V and 150 V, respectively. The voltage before the collision cell was either set to 100 V or 250–300 V, respectively, optimizing for desolvation of the intact complex or the subsequent ejection of subunits, respectively. Spectra were calibrated using an aqueous solution of cesium iodide (25 mg/ml) and exported from MassLynx. All further data analysis was performed with in-house developed python scripts (Python 3.6). When applicable, charges were assigned to charge state resolved peak series by extracting the top position for consecutive charge states and minimizing the standard deviation (SD) of the average mass by trying different charge states. Centroids for empty and cargo filled encapsulins ($T = 3$) were calculated using all data points above 40 % of the base peaks intensity in the appropriate region ($m/z$ 30,000–40,000 for empty and $m/z$ 35,000–45,000 for cargo filled encapsulins). The average was taken over three technical replicates and the error represents the

standard deviation (SD). To estimate the mass from the $m/z$ position, we fitted 77 empirical determined masses and their corresponding $m/z$ positions to the equation Mass[kDa] $= A*m/z^B$. These 77 proteins consist of encapsulin $T = 1$, which mass was determined in this study as well as 76 other assemblies, which were already measured and reported in previous publications[73,74]. The resulting formula Mass[kDa] $= 1.63*10^{-6}*m/z^{2.14}$ was used to calculate the mass from the average $m/z$ positions of empty and cargo filled encapsulins ($T = 3$). Since mass and $m/z$ positions do not follow a linear relation, we averaged the upper and lower error from the $m/z$ dimension projected in the mass domain. Cargo load was estimated based on the difference between the predicted masses of empty and cargo loaded encapsulins ($T = 3$). The error of the mass difference was calculated using the equation $\sigma\Delta$Mass $= (\sigma$Mass$1^2 + \sigma$Mass$2^2)^{1/2}$. The difference in mass was then divided by 41.4 kDa, the mass of the DD-mEos4b-EncSig monomer protein. The $A^{FLAG}$ monomer mass was calculated as the weighted average for the different proteoforms, using the summed intensities over the charge states for each species.

**Complementation of split PAmCherry1 inside encapsulins.** Cells transfected with C-PAs1 and B-PAs2 with or without $A^{FLAG}$ were seeded onto 8-well Poly-L-lysine-coated microscopy chips (Ibidi). Thirty-six hours post transfection, live cell confocal microscopy was conducted on a Leica SP5 system (Leica Microsystems). For photoactivation of split PAmCherry1, samples were illuminated with a 405 nm laser for 60 s at 40% laser power. The signal of complemented split PAmCherry1 was excited using the 561 nm laser. To quantify the complementation of split PAmCherry 1 with or without the encapsulin shell, the ratio of the total mean fluorescence after photoactivation divided by the signal before was calculated. ImageJ was used to quantify mean fluorescence values from randomly chosen areas on the well.

**Multispectral optoacoustic tomography.** Optoacoustic images of cells co-expressing $A^{STII}$ and $^{Myc}$D-BmTyr were acquired on an inVision 256-TF system (iThera Medical GmbH). Briefly, $\sim 10^7$ HEK293T cells co-expressing the genes treated with 10 µM CuCl$_2$ and 1 mM L-tyrosine 24 h prior to the measurement were detached using trypsin, washed with PBS, embedded into 1 % low melting agar yielding a tubular phantom of $\sim$ 300 µl volume. The cell phantom and an ink phantom (OD $= 0.2$) were placed in a custom-built sample holder and optoacoustic images were acquired for the range of wavelengths between 690 and 900 nm. Signals were reconstructed using ViewMSOT software suite (iThera Medical GmbH) and linearly unmixed using a reference spectrum for melanin.

**Magnetic sorting.** Cells were washed twice with PBS, detached with Accutase® (Sigma-Aldrich) and resuspended in DPBS supplemented with 10% fetal bovine serum (Gibco) prior to sorting. For magnetic sorting, columns filled with ferromagnetic spheres (MS columns, Miltenyi Biotec) were placed in an external magnetic field (OctoMACS separator, Miltenyi Biotec) and equilibrated with 1 ml DPBS containing 10% FBS. The column was loaded with cells and washed with 0.5 ml DPBS; the flow-through was collected as one fraction. After removing the column from the magnetic separator, cells were eluted with 1 ml DPBS. The total number of cells before sorting as well as the cell numbers in flow-through and eluate were determined with a Countess II FL Automated Cell Counter (Life Technologies).

**Magnetic resonance imaging of cells.** MR images were acquired at a Bruker BioSpec 94/20USR, 9.4T system equipped with a RF RES 400 1H 112/072 Quad TR AD resonator. For $T_2$* measurements of cell pellets, $4*10^6$ HEK293T cells were seeded 24 h prior to transfection on poly-L-lysine-coated 10 cm cell culture dishes. Twenty four hours post transfection, ferrous ammonium sulfate (FAS) was added to the medium yielding a concentration of 1 mM. Twenty four hours post iron addition, cells were washed three times with DPBS and detached with Accutase® and centrifuged at $500 \times g$ for 4 min. The pellets were resuspended in 800 µl DPBS and transferred to cryobank vials (Thermo Scientific Nunc) containing 50 µl of solidified 1% agarose at the bottom. Cells were then spun down at $2000 \times g$ for 2 min and immediately used for MRI. $T_2$* measurements were conducted in a custom-made holder filled with DPBS to avoid susceptibility artifacts. $T_2$* values were calculated based on a multiple gradient echo (MGE) sequence with a TR of 800 ms, 12 echoes with an echo spacing of 4.5 ms (3.5–58.5 ms), a flip angle of 50°, field of view of $65 \times 65$ mm and a matrix size of $256 \times 256$. Relaxation rates were calculated with the *Image Sequence Analysis Tool* from Bruker BioSpin MRI GmbH.

**In vivo expression of encapsulins in murine brains.** Mice were positioned in a stereotaxic frame, anesthetized with isoflurane, and implanted bilaterally with MRI compatible guide cannulae (Plastics One) that were stably fixated with dental cement. Injection cannulae (Plastics One) were connected via polyethylene tubing (PE-50), filled with silicone oil, connected to a PhD 2000 syringe pump (Harvard Apparatus) and backfilled with solutions containing AAV viral particles. Injection cannulae were inserted into the guide cannulae and lowered into the brain. A volume of 1 µl of viral particles was injected at 0.1 µl/min. Injection cannulae were subsequently retracted slowly and replaced with dummy cannulae (Plastics One) that screwed firmly into the guide cannula pedestals. All experiments on mice were conducted in accordance with the guidelines approved by the government of Upper Bavaria.

**Immunohistochemistry.** Three to six weeks after intracranial viral injection, mice were terminally anesthetized, perfused, and the brains were removed for cryosectioning. Brain slices were then blocked in SuperBlock (TBS) Blocking Buffer (Thermo Fisher Scientific) for 1 h at room temperature in a humidified chamber. Subsequently, brain slices were incubated in 5 µg/ml Monoclonal ANTI-FLAG® M2 antibody (SIGMA F1804, Sigma-Aldrich) in TBS for 2 h at room temperature. After $5 \times 5$ min washes with TBS, the brain slices were incubated in 1 µg/ml Goat anti-Mouse IgG (H + L) Cross-Adsorbed Secondary Antibody conjugated to Alexa Fluor 488 (A-11001, Invitrogen) in TBS for 1 h at room temperature in darkness. For nuclear counterstaining, DAPI was added at 300 nM for 5 min. Finally, the brains slices were washed five times with TBS. Brain slices were subsequently imaged on an EVOS FL Auto Cell Imaging System (Invitrogen) or a Leica SP 5 confocal microscope (Leica Microsystems).

**In vivo MRI.** HEK293T cells ($\sim 4*10^6$) were seeded onto poly-L-lysine-coated 10 cm cell culture dishes. Twenty four hours after seeding, cells were transiently transfected at 70–80% confluency with DNA constructs encoding either $A^{FLAG} + B^{M7}$ or $A^{FLAG}$ + mEos4b-EncSig, as well as for both conditions Zip14 at 5% of the total DNA amount using X-tremeGENE™ (Roche). Twenty four hours post transfection, the cell culture medium was replenished with fresh medium containing 1 mM FAS. Twenty four hours after incubation with FAS, cells were washed gently three times with PBS, detached from the culture dishes after 5 min of treatment with a 1:1 solution of Accutase® (Sigma) and Trypsin, centrifuged for 5 min at $1200 \times g$ and resuspended in growth media. Cell suspensions were backfilled into two injection cannulae (28 Gauge, Plastics One, Roanoke, VA, USA) connected via plastic tubing to 25 µl Hamilton glass syringes clamped in a remote dual syringe pump (PHD 22/2,000; Harvard Apparatus, Holliston, MA, USA). Injection cannulae (the side of injection for $A^{FLAG} + B^{M7}$ or control were switched between experiments) were then lowered into bilateral guide cannulae (22 Gauge, Plastics One, Roanoke, VA) that were previously implanted in Sprague–Dawley rats[75]. Rats were then centered in the bore of a 7T 20 cm inner diameter, horizontal bore magnet (Bruker BioSpin MRI GmbH, Ettlingen, Germany) and gradient echo scans (FOV $= 2.5$ cm $\times 2.5$ cm, matrix size $= 256 \times 256$; seven slices with 1 mm slice thickness) were taken at a TR $= 800$ ms and different echo times (5, 10, 15, 20, 25 ms) to compute relaxation rate maps and perform ROI analysis (circular ROIs of 1 mm diameter placed on injection sites) using custom routines in Matlab (Mathworks, Natick, MA, USA). All procedures on rats were conducted in accordance with National Institutes of Health guidelines and with the approval of the MIT Committee on Animal Care.

**Cell viability assays.** Iron-related cytotoxicity was monitored via the Roche Cytotoxicity Detection Kit (LDH) (Roche Diagnostics) according to the protocol of the manufacturer. Briefly, $7.5*10^5$ HEK293T cells were seeded on poly-L-lysine-coated 24-well plates. Twenty four hours post seeding, cells were transfected with different combinations of genes using X-tremeGENE HP (Roche). The Zip14 DNA amount was kept constant in all samples expressing Zip14 (5% of total DNA). For expression of combinations of $A^{FLAG}$ with cargo proteins, 60% of the total DNA amount was encoding $A^{FLAG}$ and the remaining 35% were used for the respective cargo molecule. 24 h post transfection, FAS was added to the medium from a 100 mM stock solution yielding a final concentration of 2.5 mM. Twenty four hours after addition of FAS, cells were assayed for LDH release. In order to evaluate gene-related toxicity in the absence of iron, the assay was performed accordingly but without iron supplementation and cells were assayed 48 h post transfection. The Luciferase-based viability assay (RealTime-Glo™ MT Cell Viability Assay, Promega) was performed according to the protocol of the manufacturer in 96-well plate format as an endpoint measurement. Luminescence readings were taken on a Centro LB 960 (Berthold Technologies) at 0.5 s acquisition time.

**Electron microscopy.** Please refer to Supplementary Methods in the Supplementary Information for a detailed description of the electron microscopy techniques used.

**Data availability.** Data are available upon reasonable request to the corresponding author. The cryo-EM maps of non-iron loaded and iron loaded encapsulins in HEK293T cells have been deposited under EMDB-4392 and 4393 respectively.

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

## Acknowledgements

We are grateful for support from the European Research Council under grant agreements ERC-StG: 311552 (F.S., A.S., H.R., G.G.W.). T.W., J.S. and A.J.R.H acknowledge support from the Netherlands Organization for Scientific Research (NWO) funding the large-scale proteomics facility Proteins@Work (project 184.032.201) embedded in the Netherlands Proteomics Centre. We acknowledge Susanne Pettinger for assistance with DLS measurements.

## Author contributions

F.S. co-designed the study, generated all constructs, conducted all cell and biochemical experiments, analyzed data, generated figures and co-wrote the manuscript; C.M. made important contributions to the iron-loading experiments and performed in vitro MRI experiments; P.E. designed, conducted and analyzed cryo-EM experiments supervised by J.P.; A.S. supported cell experiments; H.R. supported animal experiments; M.D. and S.B. conducted in vivo MRI experiments supervised by A.J.; A.G. helped with HPLC purification and DLS analysis; T.P.W. and J.S. conducted and analyzed native mass spectrometry experiments supervised by A.J.R.H., H.F. and M.H. supervised in vitro MRI experiments; V.N. supervised MSOT experiments; G.G.W. conceptualized and co-designed the study, analyzed data and generated figures, supervised the project, and wrote the manuscript.

## Additional information

**Competing interests:** The authors declare no competing interests.

