## [Peer Review File · Nature Communications]

Reviewers' comments:

Reviewer #1 (Remarks to the Author):

The manuscript "Bacterial encapsulins as orthogonal compartments for mammalian cell engineering" by Sigmund et al. describes the heterologous expression of prokaryotic self-assembling protein compartments in eukaryotic cells for the cellular engineering of spatially confined processes with applications in molecular imaging, metabolic engineering and cellular therapy. The encapsulin shell protein EncA from *Myxococcus xanthus* is shown to assemble readily into nanocompartments to which a set of native and engineered proteins is targeted. An impressive array of possible applications is demonstrated, including encapsulation of bimolecular fluorescence complementation, enzymatic reactions, gene reporters for optoacoustics, iron biomineralisation for MRI detection and cell sorting. This work is a major step towards the development of "designer organelles", not only for the demonstrated applications, but also for metabolic engineering of orthogonal reactions in eukaryotic cells. The manuscript is well written, the figures accurately presented and the claims made are novel and will be of great interest to many others in the field. As the authors noted themselves it is surprising how readily the encapsulins in this study seem to assemble and pack with cargo proteins considering the sensitivity of assembly to protein concentration and the complex cellular environment of mammalian cells compared to prokaryotic hosts.

Comments:

The authors assume that the nanocompartments assemble *in vivo* and provide a number of convincing data to support this claim e.g. Native PAGE, cryoEM images of purified encapsulins. Nevertheless, a TEM image of a eukaryotic cell containing heterologous encapsulins would very much strengthen this paper and give crucial information about the abundance and localisation of the encapsulins within the cells.

The authors talk about engineering size-controlled metal biomineralization. I thought this could potentially be miss-understood suggesting that this is about controlling the size of the metal biominerals. Whereas in this study, dose-dependent iron loading of encapsulins with a given size is described the actual size of the encapsulin is not modified.

Selective access of substrates via the pores in the nanoshell and confinement of toxic melanin is claimed. This should be phrased a bit more carefully because the selectivity of encapsulins is not well understood yet. Yes, the data has demonstrated that the small molecule tyrosine can access the compartment and indicated confinement of the toxic melanin polymer, but there is no direct proof that melanin is indeed confined in the nanocompartments.

In the introduction, the authors mention engineering compartments in prokaryotes for biotechnological applications. In this context, reference should also be made to bacterial microcompartments.

Page 2, Results: Encapsulation expression and auto-assembly

♣ Is there a rationale for choosing MycC-IntP2A-ASTII for purification from HEK293T cells and not B or D or all three proteins? Similarly, why was B chosen for *in vivo* expression work?

♣ A brief explanation of the function of natural encapsulin cargo proteins B, C, D would be useful and could be included in introduction.

♣ "Furthermore, robust secretion of StrepTagII-modified encapsulins from HEK293T cells was possible by addition of a N-terminal BM40 secretion signal as shown by Coomassie-stained BN-PAGE of material present in the cell culture supernatant (Fig. 1i)."

In the introduction, it is stated that "The termini of the shell protein extend to the inner and outer surface, respectively, such that surface functionalizations are possible." Is this a prediction based on the structure? If the BM40 secretion signal faces inwards how can it be recognized for

secretion? Please clarify.

Page 3, Results: Bioengineering melanosomes as non-toxic gene reporters for optoacoustics

♣ "Encapsulation of tyrosinase also lead to a significant reduction of cell viability as assessed by a luciferase-based viability assay." I assume you mean increase in cell viability.

Page 4, Results: Encapsulins enable detection via MRI and Magnetic Sorting

♣ "The nanocompartment AFLAG co-expressed with all native cargo or just with ferritin-like B lead to a similar increase in R^* -relaxation rates as compared to just AFLAG."

This does not correspond to Fig 5d where it is shown that:

"Expression of AFLAG + BCD P2A or MycB shows a significantly enhanced R^* -relaxation rate as compared with the nanocompartment alone or loaded with MycD."

Minor comments:

Page 1: M. xanthus, full name is required before using abbreviation

TOC figure: Chemical structures and arrows are hard to see

Figure 2c: IP anti FLAG SDS silver gel

Labelling of PAmCherry bands missing. A looks like a double band, do you have an explanation?

BN Fluor: Fluorescence hard to see

Figure 2d: confocal microscopy of PAmCherry

How is $n=3$ defined? 3 different areas of the well? Where areas or individual cells picked for fluorescent signal measurements?

Figure 2e: luminescence signal

How is $n=3$ defined?

Figure 2g: on-gel formation of CdS nanodots

Band on gel is hard to see

Figure 3c:

How is $n=3$ defined?

Supplementary Figure 3b:

Labelling for 'non-transfected cells' is missing

Reviewer #2 (Remarks to the Author):

In this report encapsulins are heterologously expressed as nanoreactors in mammalian cells. They serve as containers for a number of enzymes, and are used for iron storage. The latter system is also employed in vivo to detect specific xenografted cells in rat brain. Although encapsulins have been known as protein-based nanoreactors, their stable genetic introduction in eukaryotic cells is novel and conceptually an important step forward. The body of work presented is impressive and makes this paper of sufficient quality, depth and novelty to meet the Nature Comm standards. I don't have many comments to make. The only thing the authors should improve is quantification. The proteins expressed should be analyzed by mass spectrometry to validate their structure.

Furthermore, the nanoreactors made should be analyzed by a SEC or asymmetric field flow fractionation method coupled to MALLs to identify the exact mass of the nanoreactors. This would also allow the degree of loading of the nanoreactors. This should be at least performed for a selected number of particles that are excreted or purified via affinity chromatography. The >1.2 MDa indication is too qualitative.

Reviewer #3 (Remarks to the Author):

This is an excellent paper on using encapsulins as cell-engineering tools, with a remarkable breadth of experimental reporter systems. I thoroughly enjoyed reading this paper and would be pleased to see it published with minor revisions for clarity and to expand on some of the methods and analyses for reproducibility.

The major conclusions are well justified by the data, however some of the results need further clarification and additional supplementary evidence.

Fig1: It would be useful if the authors could provide details in the methods, or figure legend as to how the molecular weight approximations for the BN-PAGE experiments were reached. Some later figures show the standards, but in panels d/g and i these are not present.

Panel f of Fig1 shows a section of a cryo-electron micrograph. For full disclosure and to ensure the sample is representative of the sample as a whole, it would be helpful for readers to see a full-frame micrograph, provided as a supplementary figure. For a more rigorous measurement, an analysis of the measurement >100 particles and a size-distribution histogram would be useful too.

Fig5: Panel e, the results of the MRI R2* relaxation experiments are really not convincingly significant. The description of these in the results as 'showing a trend' towards elevated relaxation rates with a $p=0.0834$ is not justified. I would recommend removing this section, as the data are only suggestive and not truly significant.

In the discussion section the claim that iron-loading of heterologously expressed encapsulins in *E. coli* has not been demonstrated is not justified. The 2017 Nature Microbiology paper by Giessen and Silver clearly shows iron loading of the IMEF encapsulin system in *E. coli* (<https://www.nature.com/articles/nmicrobiol201729/figures/5>). Please adjust the manuscript accordingly.

In the methods section for the Electron Microscopy it is not clear how the protein used was purified. The section on isolation of the recombinant encapsulins is rather sparse and lacks detail beyond cell lysis. Were the encapsulins used for the EM further purified by FLAG-affinity purification, or were they used as is? Please add detail in these sections.

General Reply to all Reviewers

We thank the reviewers for their very constructive feedback on our manuscript. We respond to each of the comments in a point-by-point fashion below with cross-references to the new and updated figures.

To incorporate the new data on the native mass and cargo loading of the heterologously expressed encapsulins, we have swapped the order of the original **Figures 2 and 3** and complemented them with the new **Supplementary Figures 2 and 3**.

We have also added *in situ* cryo-EM data in a new **Figure 6 and Supplementary Figure 6**.

A list of references is given at the end of the document.

Reviewers' comments:

Reviewer #1 (Remarks to the Author):

The manuscript “Bacterial encapsulins as orthogonal compartments for mammalian cell engineering” by Sigmund et al. describes the heterologous expression of prokaryotic self-assembling protein compartments in eukaryotic cells for the cellular engineering of spatially confined processes with applications in molecular imaging, metabolic engineering and cellular therapy. The encapsulin shell protein EncA from *Myxococcus xanthus* is shown to assemble readily into nanocompartments to which a set of native and engineered proteins is targeted. An impressive array of possible applications is demonstrated, including encapsulation of bimolecular fluorescence complementation, enzymatic reactions, gene reporters for optoacoustics, iron biomineralisation for MRI detection and cell sorting. This work is a major step towards the development of “designer organelles”, not only for the demonstrated applications, but also for metabolic engineering of orthogonal reactions in eukaryotic cells.

The manuscript is well written, the figures accurately presented and the claims made are novel and will be of great interest to many others in the field. As the authors noted themselves it is surprising how readily the encapsulins in this study seem to assemble and pack with cargo proteins considering the sensitivity of assembly to protein concentration and the complex cellular environment of mammalian cells compared to prokaryotic hosts.

Comments:

R1C1: The authors assume that the nanocompartments assemble *in vivo* and provide a number of convincing data to support this claim e.g. Native PAGE, cryoEM images of purified encapsulins. Nevertheless, a TEM image of a eukaryotic cell containing heterologous encapsulins would very much strengthen this paper and give crucial information about the abundance and localisation of the encapsulins within the cells.

Response to R1C1: We have now added cryo-EM data from lamella FIB-milled from HEK293T cells stably expressing encapsulins (new **Figure 6**). The histogram of the interparticle distances

shown in **Supplementary Figure 6** demonstrates the uniform, monodisperse, and cytosolic expression of cargo-filled encapsulins.

R1C2: The authors talk about engineering size-controlled metal biomineralization. I thought this could potentially be miss-understood suggesting that this is about controlling the size of the metal biominerals. Whereas in this study, dose-dependent iron loading of encapsulins with a given size is described the actual size of the encapsulin is not modified.

Response to R1C2: We thank the reviewer for this comment regarding our terminology. Since the term 'size-controlled' seems to be indeed used mainly in the context of synthesizing differently sized structures, we now use the expression 'size-constrained biomineralization' to indicate that the single type of encapsulin shell that we use in this study constrains the size of iron biomineralization.

R1C3: Selective access of substrates via the pores in the nanoshell and confinement of toxic melanin is claimed. This should be phrased a bit more careful because the selectivity of encapsulins is not well understood yet. Yes, the data has demonstrated that the small molecule tyrosine can access the compartment and indicated confinement of the toxic melanin polymer, but there is no direct proof that melanin is indeed confined in the nanocompartments.

Response to R1C3: As the reviewer pointed out, our data jointly support that melanin can be produced by the encapsulated tyrosinase inside the encapsulin compartment which entails transport of tyrosine into the encapsulins. Although the compartmentalized melanin production strongly reduces melanin-mediated toxicity, the reviewer correctly points out that since melanin is not precisely defined in size, it is non-trivial to assess which exact 'melanin polymers' can or can not exit the encapsulin shell. We thus wrote 'compartmentalized melanin-production' and not 'spatially confined melanin' or similar. We have now also changed the term 'melanin-filled encapsulins' to 'melanin producing encapsulins' (p. 3, legend for **Figure 2**).

R1C4: In the introduction, the authors mention engineering compartments in prokaryotes for biotechnological applications. In this context, reference should also be made to bacterial microcompartments.

Response to R1C4: We have now added also references to engineered bacterial microcompartments such as Eut compartments and carboxysomes in the introduction (p. 1).

R1C5: Page 2, Results: Encapsulation expression and auto-assembly
Is there a rationale for choosing MycC-IntP2A-ASTII for purification from HEK293T cells and not B or D or all three proteins? Similarly, why was B chosen for in vivo expression work?

Response to R1C5: The scarless MycC-IntP2A-ASTII construct yielded very stable encapsulin nanocompartments, and their purification at a medium scale was efficient. B was chosen as a cargo molecule for the *in vivo* expression because our *in vitro* experiments showed that co-expression of just B was sufficient for iron loading and because the corresponding P2A construct was small enough to be efficiently packed into AAV5.

R1C6: A brief explanation of the function of natural encapsulin cargo proteins B, C, D would be useful and could be included in introduction.

Response to R1C6: We thank the reviewer for requesting this background information that can certainly be interesting to the reader. We added a paragraph on what is known about the function of the natural cargo proteins B, C, and D in the introduction (bottom of p. 1).

R1C7: “Furthermore, robust secretion of StrepTagII-modified encapsulins from HEK293T cells was possible by addition of a N-terminal BM40 secretion signal as shown by Coomassie-stained BN-PAGE of material present in the cell culture supernatant (Fig. 1i).”

In the introduction, it is stated that “The termini of the shell protein extend to the inner and outer surface, respectively, such that surface functionalizations are possible.” Is this a prediction based on the structure? If the BM40 secretion signal faces inwards how can it be recognized for secretion? Please clarify.

Response to R1C7: The human BM40 (osteonectin SPARC) secretory signal peptide (SP) marks proteins for the entry into the secretory pathway and has previously been used to secrete proteins from HEK293 cells (Holden et al. 2005). This peptide will first be recognized by the SRP in the ER membrane and accordingly translocated via the Sec61 complex into the ER lumen (Rapoport 2007), where the signal peptide is cleaved off by signal peptidases present in the ER membrane, after which encapsulin assembly most likely takes place. Subsequently, the protein assembly is transported via the Golgi network to the plasma membrane, where the Golgi-derived vesicles will fuse and release the assembled encapsulins into the extracellular space. We have added a sentence to the manuscript to explain this detail (p. 2).

R1C8: Page 3, Results: Bioengineering melanosomes as non-toxic gene reporters for optoacoustics

“Encapsulation of tyrosinase also lead to a significant reduction of cell viability as assessed by a luciferase-based viability assay.” I assume you mean increase in cell viability.

Response to R1C8: Thank you for pointing out this mistake that we have now corrected.

R1C9: Page 4, Results: Encapsulins enable detection via MRI and Magnetic Sorting

“The nanocompartment AFLAG co-expressed with all native cargo or just with ferritin-like B lead to a similar increase in R^* -relaxation rates as compared to just AFLAG.”

This does not correspond to Fig 5d where it is shown that:

“Expression of AFLAG + BCD P2A or MycB shows a significantly enhanced R_2^* -relaxation rate as compared with the nanocompartment alone or loaded with MycD.”

Response to R1C9: The observation is that co-expression of BCD or just B together with the shell A^{FLAG} both lead to significant increases in the relaxation rate. We have modified the respective sentence in the main text to make this more clear:

“The nanocompartment AFLAG co-expressed with all native cargo proteins (BCD) lead to an increase in R_2^* -relaxation rates as compared to just AFLAG. The same effect was observed by co-expressing just the ferritin-like B (**Fig. 5d**, $n=7$ from 4 independent experiments, Kruskal-Wallis, $p=0.0047$, Dunn’s nonparametric comparison vs. AFLAG at $\alpha=0.05$).”

Minor comments:

R1C10: Page 1: M. xanthus, full name is required before using abbreviation

Response to R1C10: Thank you for pointing this out. We have now added the full name *Myxococcus xanthus* in the introduction.

R1C11: TOC figure: Chemical structures and arrows are hard to see

Response to R1C11: We have improved the resolution of the TOC figure.

R1C12: Figure 2c: IP anti FLAG SDS silver gel

Labelling of PAmCherry bands missing. A looks like a double band, do you have an explanation?

BN Fluor: Fluorescence hard to see

Response to R1C12: We have now indeed identified the A^{FLAG} band as double band that is only visible when a low protein amount is applied onto the SDS-gel and subsequently silver-stained. When higher amounts are applied, the bands merge since the two species have a very similar electrophoretic mobility.

Native mass spectrometry of the encapsulins purified from HEK293T cells revealed an 80 Da mass difference with respect to the calculated mass when single subunits were ejected from the native complex. This mass difference is characteristic for phosphate groups (Nita-Lazar, Saito-Benz, and White 2008). Therefore we purified FLAG-tagged encapsulins and dephosphorylated them with calf intestinal phosphatase (CIP). Subsequent SDS-PAGE analysis

revealed that the upper band disappeared and only the lower non-phosphorylated band remained (**Supplementary Figure 3c-e**) indicating that a certain extent of the A subunits in the assembled nanocompartment are phosphorylated.

We have now also repeated the BN-PAGE of cells expressing encapsulins filled with the two PAmCherry1 split parts and exposed the gel for a longer duration on the UV imager yielding a fluorescent band with better visibility (now **Figure 3b**).

R1C13: Figure 2d: confocal microscopy of PAmCherry

How is n=3 defined? 3 different areas of the well? Where areas or individual cells picked for fluorescent signal measurements?

Response to R1C13: Three independent transfection experiments were conducted (biological replicates). Furthermore, we averaged the mean fluorescence over 3 random areas on the plate for each biological replicate. We have updated the figure legend accordingly (now **Figure 3c**).

R1C14: Figure 2e: luminescence signal

How is n=3 defined?

Response to R1C14: The experiment was conducted 3 times with independent transfections (biological replicates) and the mean luminescence signal (obtained from 3 technical replicates) was averaged across the biological replicates. We have updated the figure legend accordingly (now **Figure 3d**).

R1C15: Figure 2g: on-gel formation of CdS nanodots

Band on gel is hard to see

Response to R1C15: We repeated the experiment and exposed the gel for a longer duration to obtain better visibility of the band (now **Figure 3f**).

R1C16: Figure 3c:

How is n=3 defined?

Response to R1C16: The experiment was conducted 3 times in biological replicates (individual transfection experiments). In each biological replicate an average over 3 random plate positions was calculated. This panel is now shown in **Figure 2d**.

R1C17: Supplementary Figure 3b:

Labelling for 'non-transfected cells' is missing

Response to R1C17: We have clarified the label in the legend of the Figure which is now **Supplementary Figure 5**.

Reviewer #2 (Remarks to the Author):

In this report encapsulins are heterologously expressed as nanoreactors in mammalian cells. They serve as containers for a number of enzymes, and are used for iron storage. The latter system is also employed *in vivo* to detect specific xenografted cells in rat brain. Although encapsulins have been known as protein-based nanoreactors, their stable genetic introduction in eukaryotic cells is novel and conceptually an important step forward. The body of work presented is impressive and makes this paper of sufficient quality, depth and novelty to meet the Nature Comm standards.

R2C1: I don't have many comments to make. The only thing the authors should improve is quantification. The proteins expressed should be analyzed by mass spectrometry to validate their structure.

Furthermore, the nanoreactors made should be analyzed by a SEC or asymmetric field flow fractionation method coupled to MALLs to identify the exact mass of the nanoreactors. This would also allow the degree of loading of the nanoreactors. This should be at least performed for a selected number of particles that are excreted or purified via affinity chromatography. The >1.2 MDa indication is too qualitative.

Response to R2C1:

We have conducted the additional analyses that the reviewer requested and describe them here in the order in which the reviewer has mentioned each point.

With respect to the structure of the encapsulins, we have now added cryo-EM data from cells showing encapsulins loaded with all endogenous cargo molecules (BCD) as monodisperse particles with a narrow size distribution of 31.1 ± 0.1 nm (mean \pm SEM, $n = 298$) consistent with the 180-mer assembly with T=3 symmetry (**new Figure 6 and new Supplementary Figure 6**). These measures were obtained from individual 2x binned subtomograms with a pixel size of 1.368 nm thus accounting for the difference to 32 nm that was given as diameter in the literature for encapsulins expressed in prokaryotes (McHugh et al. 2014).

This size distribution is comparable to the one we obtained by cryo-EM from purified encapsulins loaded with just the native cargo C (**Figure 1**) for which we have now added the size distribution (peak of the histogram is at a diameter of ~32 nm, **new Supplementary Figure 1a,b**). In this sample, we also observed a smaller species with a diameter of ~18 nm which is in agreement with cryo-EM data showing particles consistent with T=1 symmetry observed upon expression of *M. xanthus* encapsulin in *E.coli* (McHugh et al. 2014).

We have complemented these cryo-EM data with native Mass Spectrometry data from encapsulins purified from HEK cells using a QToF-2 mass spectrometer modified for the analysis of high-mass ions. For encapsulins (A^{FLAG}) expressed without cargo proteins, we resolved one charge state distribution around m/z 17,000, whose calculated mass corresponds very well to the 60-mer with T=1 symmetry. We also observed an unresolved charge state

distribution around m/z 33,000. For encapsulins loaded with a fluorescent protein as cargo (mEos4b), we observed only one unresolved charge state around m/z 38,000. These data are consistent with BN-PAGE data obtained from the identical samples, which also showed the presence of a smaller band (**new Supplementary Figure 2c**). Since no charge state could be resolved, which may also be due to the heterogeneity in the monomer mass, we predicted the mass based on the m/z position utilizing empirical data of 77 protein assemblies (Snijder et al. 2013; Veessler et al. 2014), **new Supplementary Figure 3a**).

In addition, we have analyzed the same samples (unloaded and cargo-loaded encapsulins) by size exclusion chromatography (SEC) using a Superose® 6 10/300 GL connected to an Äkta FPLC. This column seemed the most suitable of the commercially available columns although its separation range with 5 - 5000 kDa is not ideal for the sizes at hand. SEC analysis showed that the encapsulins expressed without cargo showed an additional peak in the refractive index readings, again indicating the presence of a less abundant smaller species (**new Supplementary Figure 2a**) which is also clearly visible on BN-PAGE (**new Supplementary Figure 2c**). Whereas we could not directly estimate the mass from SLS during SEC because of the strong scattering, we performed DLS on the same purified material and obtained an estimated hydrodynamic radius of 22.7 ± 0.35 and 21.1 ± 0.63 nm and (mean and SD over several dilutions from each sample for two independent samples) for A+DD-mEos4B-EncTag. This hydrodynamic radius is larger than the size determined by cryo-EM which is a common observation that can, e.g., be due to the hydration layer not visible by EM (Bootz et al. 2004) and is consistent with previous measurements obtained from encapsulins (Rahmanpour and Bugg 2013; Tamura et al. 2015). Just the purified A^{FLAG} encapsulin shell yielded an average hydrodynamic radius of 17.1 ± 0.11 nm and 19 ± 1.15 nm from 2 independent samples which again is consistent with the presence of a smaller species in the absence of cargo molecules. We have added a point in the discussion on the apparent role of the cargo in favoring assembly of the 180-mer encapsulin with T=3 symmetry.

We furthermore estimated the degree of cargo loading based on the difference between the predicted masses of empty and cargo loaded encapsulins. In both cases, we estimated the cargo load by dividing the mass difference by 41.4 kDa, the mass of the DD-mEos4b-EncTag monomer, and arrived at an average of 62 ± 8 cargo molecules per nanocompartment which was confirmed by gel densitometry which gave an estimate of 63 cargo molecules on average (**new Supplementary Figure 3a,b**).

We thank the reviewer for asking us to conduct these additional characterizations that will be helpful for future applications of encapsulins in mammalian cells.

Reviewer #3 (Remarks to the Author):

This is an excellent paper on using encapsulins as cell-engineering tools, with a remarkable breadth of experimental reporter systems. I thoroughly enjoyed reading this paper and would be pleased to see it published with minor revisions for clarity and to expand on some of the methods and analyses for reproducibility.

The major conclusions are well justified by the data, however some of the results need further clarification and additional supplementary evidence.

R3C1: Fig1: It would be useful if the authors could provide details in the methods, or figure legend as to how the molecular weight approximations for the BN-PAGE experiments were reached. Some later figures show the standards, but in panels d/g and i these are not present.

Response to R3C1: We have now added information about the commercially available NativeMark™ Unstained Protein Standard (Life Technologies) for BN-PAGE covering a size range between 20 and 1200 kDa in the Methods section (**Blue Native gel electrophoresis and on-gel analyses**). We have omitted the Marker in d/g and I because of limited space but please see new **Supplementary Figure 2 and 4** that display gels showing the marker bands side-by-side. The electrophoretic mobility of the encapsulin bands in relation to the marker shown in these figures was consistently observed in all BN-PAGE experiments. Please also see the additional characterization of the heterologously expressed encapsulins by native mass spectrometry in the new **Supplementary Figures 2 and 3**.

R3C2: Panel f of Fig1 shows a section of a cryo-electron micrograph. For full disclosure and to ensure the sample is representative of the sample as a whole, it would be helpful for readers to see a full-frame micrograph, provided as a supplementary figure. For a more rigorous measurement, an analysis of the measurement >100 particles and a size-distribution histogram would be useful too.

Response to R3C2: We have now added a size distribution histogram of 492 particles and a set of representative particle images to the supplementary information (**Supplementary Figure 1a,b**). Please also see the in situ cryo-EM data in the new **Figures 6 and Supplementary Figure 6**.

R3C3: Fig5: Panel e, the results of the MRI R2* relaxation experiments are really not convincingly significant. The description of these in the results as 'showing a trend' towards elevated relaxation rates with a $p=0.0834$ is not justified. I would recommend removing this section, as the data are only suggestive and not truly significant.

Response to R3C3: We have now added 5 new datasets and obtain a *p*-value of $p=0.0084$ (paired t-test, $n=9$) from an ROI analysis of the R_2 maps.

R3C4: In the discussion section the claim that iron-loading of heterologously expressed encapsulins in *E. coli* has not been demonstrated is not justified. The 2017 Nature Microbiology paper by Giessen and Silver clearly shows iron loading of the IMEF encapsulin system in *E. coli* (<https://www.nature.com/articles/nmicrobiol201729/figures/5>). Please adjust the manuscript accordingly.

Response to R3C4: We now added a reference to the paper mentioned that shows paracrystalline electron dense structures in *E.coli* heterologously expressing encapsulins that are probably inside the lumen of the encapsulins. We write: “Whereas encapsulins heterologously expressed in *E. coli* were shown to load iron⁵⁸, iron loading of encapsulins in mammalian cells has so far not been shown.”

R3C5: In the methods section for the Electron Microscopy it is not clear how the protein used was purified. The section on isolation of the recombinant encapsulins is rather sparse and lacks detail beyond cell lysis. Were the encapsulins used for the EM further purified by FLAG-affinity purification, or were they used as is? Please add detail in these sections.

Response to R3C5: We have now added an extended section on how the protein for EM analysis was purified in the **Methods** section:

“For single particle cryoEM analysis, we co-expressed StrepTagII-tagged encapsulin shell with the Myc-tagged cargo protein C using the co-expression construct ^{Myc}C-IntP2A-A^{STII}. The StrepTagII-Streptactin system was then used for convenient purification of encapsulins from mammalian cells at a medium scale. Briefly, $\sim 10^8$ transfected HEK293T cells were washed with PBS and scraped off cell culture flasks 72 h post-transfection and thoroughly resuspended in 10 ml Buffer W (150 mM NaCl, 100 mM Tris-Cl, pH 8.0) containing protease inhibitor cocktail. Accordingly, the cells were lysed by freeze-thaw-cycling between liquid nitrogen and water at room temperature (4 cycles). Cell debris were spun down for 15 min at 10,000 x g at 4°C. The cleared supernatant was applied onto a 1 ml Gravity flow Strep-Tactin®XT Superflow® column (IBA Lifesciences). The column was washed with 5 column volumes and protein was eluted with 1.6 column volumes of Buffer BXT (150 mM NaCl, 100 mM Tris-Cl, pH 8.0, 50 mM Biotin). Before further processing, the purified protein solution was filtered through a 0.45 μ m pore filter.”

References

- Bootz, Alexander, Vitali Vogel, Dieter Schubert, and Jörg Kreuter. 2004. "Comparison of Scanning Electron Microscopy, Dynamic Light Scattering and Analytical Ultracentrifugation for the Sizing of Poly(butyl Cyanoacrylate) Nanoparticles." *European Journal of Pharmaceutics and Biopharmaceutics: Official Journal of Arbeitsgemeinschaft Fur Pharmazeutische Verfahrenstechnik e.V* 57 (2): 369–75.
- Giessen, Tobias W., and Pamela A. Silver. 2017. "Widespread Distribution of Encapsulin Nanocompartments Reveals Functional Diversity." *Nature Microbiology* 2 (March): 17029.
- Holden, Paul, Douglas R. Keene, Gregory P. Lunstrum, Hans Peter Bächinger, and William A. Horton. 2005. "Secretion of Cartilage Oligomeric Matrix Protein Is Affected by the Signal Peptide." *The Journal of Biological Chemistry* 280 (17): 17172–79.
- McHugh, Colleen A., Juan Fontana, Daniel Nemecek, Naiqian Cheng, Anastasia A. Aksyuk, J. Bernard Heymann, Dennis C. Winkler, et al. 2014. "A Virus Capsid-like Nanocompartment That Stores Iron and Protects Bacteria from Oxidative Stress." *The EMBO Journal* 33 (17). EMBO Press: 1896–1911.
- Nita-Lazar, Aleksandra, Hideshiro Saito-Benz, and Forest M. White. 2008. "Quantitative Phosphoproteomics by Mass Spectrometry: Past, Present, and Future." *Proteomics* 8 (21): 4433–43.
- Rahmanpour, Rahman, and Timothy D. H. Bugg. 2013. "Assembly in Vitro of Rhodococcus Jostii RHA1 Encapsulin and Peroxidase DypB to Form a Nanocompartment." *The FEBS Journal* 280 (9): 2097–2104.
- Rapoport, Tom A. 2007. "Protein Translocation across the Eukaryotic Endoplasmic Reticulum and Bacterial Plasma Membranes." *Nature* 450 (7170): 663–69.
- Snijder, Joost, Rebecca J. Rose, David Veessler, John E. Johnson, and Albert J. R. Heck. 2013. "Studying 18 MDa Virus Assemblies with Native Mass Spectrometry." *Angewandte Chemie* 52 (14): 4020–23.
- Tamura, Akio, Yosuke Fukutani, Taku Takami, Motoko Fujii, Yuki Nakaguchi, Yoshihiko Murakami, Keiichi Noguchi, Masafumi Yohda, and Masafumi Odaka. 2015. "Packaging Guest Proteins into the Encapsulin Nanocompartment from Rhodococcus Erythropolis N771." *Biotechnology and Bioengineering* 112 (1): 13–20.
- Veessler, David, Reza Khayat, Srinath Krishnamurthy, Joost Snijder, Rick K. Huang, Albert J. R. Heck, Ganesh S. Anand, and John E. Johnson. 2014. "Architecture of a dsDNA Viral Capsid in Complex with Its Maturation Protease." *Structure* 22 (2): 230–37.

REVIEWERS' COMMENTS:

Reviewer #1 (Remarks to the Author):

Thank you to the authors for the detailed response letter. The authors have fully addressed all comments and most notably have provided compelling new cryo EM data, strengthening the manuscript even further. Its a fantastic piece of work ready for publication.

Reviewer #2 (Remarks to the Author):

the authors have done a great job in answering the questions raised by the referees. The paper is now suitable for publication

Reviewer #3 (Remarks to the Author):

We thank the authors for their consideration of our review comments and appreciate the extra data and descriptions provided in the text. This article is now suitable for publication and we look forward to seeing it 'in print'.

General Reply to all Reviewers - 2

We thank the reviewers again for their constructive feedback and their very positive response to our revised manuscript.

Please note that based on the recommendations in the Life Science Checklist, we now chose a non-parametric test to analyze the *in vivo* MRI results, which changes the *p*-value given in the response R3C3 from $p=0.0084$ (paired t-test, $n=9$) to $p=0.0078$ (Wilcoxon matched-pairs signed rank test, $n=9$).

Please also note that we have now decided to abbreviate the minimal encapsulation signal as EncSig (as opposed to EncTag) to differentiate it better from the epitope tags in the constructs.

Thank you again for your very good feedback that helped to further improve the manuscript.